


# Features in air ions measured by an Air Ion Spectrometer (AIS) at Dome C

Xuemeng Chen[1], Aki Virkkula[1,2], Veli-Matti Kerminen[1], Hanna E. Manninen[1,3], Maurizio Busetto[4],

Christian Lanconelli[4], Angelo Lupi[4], Vito Vitale[4], Massimo Del Guasta[5], Paolo Grigioni[6], Riitta

Väänänen[1], Ella-Maria Duplissy[1], Tuukka Petäjä[1], and Markku Kulmala[1]

[1]Department of Physics, University of Helsinki, P.O. Box 64, FI-00014 Helsinki, Finland
[2]Finnish Meteorological Institute, Air Quality Research, P.O. Box 503, FI-00101 Helsinki, Finland
[3]CERN, CH-1211 Geneva, Switzerland
[4]Institute of Atmospheric Sciences and Climate, Italian National Research Council, 40129 Bologna,
Italy
[5]Istituto Nazionale di Ottica, INO-CNR, I-50019 Sesto Fiorentino (FI), Italy
[6]ENEA Laboratory for Observations and Analyses of Earth and Climate, C.R. Casaccia, I-00123
S.Maria di Galeria (RM), Italy

*Correspondence to:* Xuemeng Chen (xuemeng.chen@helsinki.fi)

**Abstract.** An Air Ion Spectrometer (AIS) was deployed for the first time at the Concordia station at
Dome C (75°06' S, 123°23' E; 3220 m above sea level), Antarctica during 22 December 2010 –16
November 2011 for measuring the number size distribution of air ions. In this work, we present results
obtained from this air ion dataset together with aerosol particle and meteorological data. The main
processes that modify the number size distribution of air ions during the measurement period at this
high-altitude site included new particle formation (NPF, observed on 85 days), wind-induced ion
formation (observed on 36 days), and ion production and loss associated with cloud/fog formation
(observed on 2 days). On days without observations of the foregoing processes or other anomalies, i.e.
event-free days, the concentration of cluster ions (0.9-1.9 nm) exhibited a clear seasonality, with high
concentrations in the warm months and low concentrations in the cold. Compared to event-free days,
days with NPF were observed with higher cluster ion concentrations. A number of NPF events were
observed with restricted growth below 10 nm, which were termed as suppressed NPF. There was



another distinct feature, namely a simultaneous presence of two or three separate NPF and subsequent growth events, which were named as multi-mode NPF events. Growth rates (GRs) were determined using two methods: the appearance time method and the mode fitting method. The former method seemed to work better for the NPF events characterized by a fast particle GR, whereas the latter method

usually worked better when the GR was slow. The formation rate of 2-nm positive ions ($J_2^+$) was calculated for all the NPF events for which a GR in the 2-3 nm size range could be determined. On average, $J_2^+$ was about 0.014 cm$^{-3}$s$^{-1}$. The ion production in relation to cloud/fog formation in the size range of 8-42 nm seemed to be a unique feature at Dome C. These ions may, however, either be multiply charged particles but detected as singly charged in the AIS, or be produced inside the

instrument, due to the cleavage of cloud condensation nuclei (CCN), possibly related to the instrumental behaviour under the extremely cold condition. For the wind-induced ion formation, our observation suggested that the ions likely from came more atmospheric nucleation of vapours released from the snow than from mechanical charging of shattered snow flakes and ice crystals.

**1. Introduction**

Air ions, also known as atmospheric ions, are electric charge carriers in the atmosphere, ranging from primary ions to charged aerosol particles (e.g. Harrison and Tammet, 2008; Hirsikko et al., 2011; Chen et al., 2016). The scientific interest in air ions has lasted for over a century and is still on-going. Air ions have a primary role in the discipline of atmospheric electricity, because their motion in the atmosphere

serves the air conductivity (Wilson, 1924; Israël, 1970; Tinsley, 2008). Air ions have also raised the interest of aerosol scientists because of their participation in atmospheric aerosol formation and thus their influences on air quality, human health and climate (Gunn, 1954; Bricard et al., 1968; Hõrrak et al., 1998; Yu and Turco, 2008; Manninen et al., 2010; Waring and Siegel, 2011).

Upon the ionisation of air molecules by ionising radiation, electric charges are created. After undergoing a series of chemical and dynamic processes with trace gases and pre-existing aerosol





particles, electric charges that survive the initial recombination and other loss mechanisms are stabilised in the form of charged nanoparticles (Chen et al., 2016). Such charged nanoparticles, i.e. stable air ions, are typically observed during new particle formation (NPF) events (Manninen et al., 2009; Hirsikko et al., 2011; Leino et al., 2016). NPF is an important source of atmospheric aerosol particles (Kulmala et

al., 2004; Poschl, 2005) and, under favourable conditions, of cloud condensation nuclei (CCN; (Kerminen et al., 2012; Dunne et al., 2016)). By this way, aerosol particles originating from NPF have a potential to influence many cloud properties and thus climate (Boucher et al., 2013; Dunne et al., 2016).

Carslaw et al. (2013) suggested that aerosol-related uncertainties in global models could be best reduced

through the study of natural aerosols in environments with negligible anthropogenic influence. Antarctica is such an environment. Long-term time series of particle number concentrations have been published both from the coastal Antarctica, including the Neumayer station (Weller et al., 2011), and from the upper plateau including the South Pole (e.g. Samson et al., 1990). Number size distributions of aerosol particles have been measured during short-term campaigns, mainly at coastal stations (e.g., Ito,

1993; Koponen, 2003; Virkkula et al., 2007; Asmi et al., 2010; Pant et al., 2011; Belosi et al., 2012; Kyrö et al., 2013; Weller et al., 2015), but also on the upper plateau at the South Pole (e.g., Park et al., 2004). Hara et al. (2011) presented particle number size distributions measured on the coast of Queen Maud Land at the Japanese station Syowa in 2003–2005. At the Norwegian Troll station in the inner region of Queen Maud Land, particle number size distributions in the size range 30 – 800 nm have been

measured over several years (Fiebig et al., 2014).

Järvinen et al. (2013) presented a 2-year-long record of particle number size distributions in the size range of 10 – 600 nm measured at the Italian-French Concordia station at Dome C on the upper Antarctic plateau, and observed clear signs of atmospheric NPF. However, since their measurement size

range started from 10 nm of particle diameter, it was not possible to get any information on the initial step of the NPF, which is expected to take place at diameters below 2 nm (Kulmala et al., 2013). Especially, the question on the role of air ions in the NPF remained totally open. Air ion number size distributions in the size range from < 1 nm up to about 40 nm have been measured at Aboa in the



coastal Antarctica during several summer campaigns (Virkkula et al., 2007; Asmi et al., 2010; Kyrö et al., 2013), but not on the upper plateau. The high altitude of Dome C makes the Concordia station more exposed to cosmic radiation than the coastal sites. Thus, more pronounced ionisation of air molecules could be expected. However, the inland location of Dome C represents a pristine environment with very limited source of vapours essential for clustering and subsequent nanoparticle formation. Therefore, it is worth investigating the synergic impact of high ionisation and low precursor vapours on the properties of air ions at this Antarctic site.

In this work, we present the first set of results on the air ion observation at Dome C. Our aims are to characterise the key features of air ions at this Antarctic site, including the seasonality of their concentrations, and to analyse the variability of air ions in relation to NPF. Moreover, the particle growth during NPF processes is studied in terms of growth rates (GRs) using two methods: the appearance time method (Lehtipalo et al., 2014) and the mode fitting method (Dal Maso et al., 2005). A comparison of GRs determined using these two methods is carried out.

## 2. Methods

The analyses in this work were based on ambient data collected from the Concordia station (75°06' S, 123°23' E) at Dome C in Antarctica during 22 December 2010 – 16 November 2011. The station is located on the Antarctic Plateau with an altitude of 3220 m above sea level and a minimum distance of 1100 km from the coastline (Becagli et al., 2012). All the data are presented in UTC.

### 2.1. Air ion and total aerosol particle measurements

#### 2.1.1. Air ion measurement

The number size distribution of air ions was measured with an Air Ion Spectrometer (AIS) during the campaign period. The AIS employs two cylindrical multi-channel aspiration-type analysers and a high sample flowrate. Such design enables it to measure negative and positive ions simultaneously down to



sizes of below 1 nm (Mirme et al., 2007).

The air sampled into the AIS is split into two equal streams. On the way to be directed to the subsequent analyser, each stream passes through a sample preconditioner, i.e. a corona charger coupled with an

electrical filter. Sample preconditioners are turned on only during the measurement of background signals, where corona chargers produce charger ions of an opposite polarity to the subsequent analysers, so that clusters and particles in each sample stream are either neutralized or assigned an opposite of polarity to the analyser and therefore generate no signal in the detection system. During the campaign, each AIS measurement cycle was composed of a 1-min background probing and a 4-min ambient

sampling. Preconditioners are off for ambient sampling. Sample streams pass directly into the respective analysers, where air ions are segregated based on their electrical mobility into different channels. The analyser used in the AIS is a variant of the differential mobility analyser (DMA). Unlike common cylindrical DMAs, where ions are collected at the inner electrode and via altering the voltage applied on the electrodes, ions of different mobility are measured (Hinds, 1999), the outer electrode in

the AIS analyser serves the collecting role and ions of different mobility are collected simultaneously by different channels. The operation of the AIS analyser is based on electrical repulsion. Therefrom, the sample flow is introduced near the inner electrode and sheath flow near the outer one. The outer electrode of the AIS analyser is divided into 21 insulated sections, each of which is connected to an electrometer as the detector. Coupling the outer electrode design with a specially shaped inner electrode,

which comprises several cylindrical sections biased at different potentials, the analyser is able to perform a concurrent classification of ions into the 21 measuring channels. More detailed technical descriptions of the instrument are presented by Mirme et al. (2007) and Mirme and Mirme (2013).

The AIS assumes the normal temperature and pressure (NTP) condition and has a total sample flowrate

of 60 l/min and a sheath flowrate of 60 l/min in each analyser at NTP. A single blower controls the whole flow system. Although sample, sheath and total exhaust flowrates are monitored by venturi flowmeters for balancing the flow system in the AIS, only the total exhaust flowrate, equivalent to the total sample flowrate, is recorded. Owing to the distinct ambient condition at Dome C from NTP, a flow





correction was applied to the recorded data to retrieve the actual number concentration of ions. In a venturi system, the volumetric flowrate ($Q$) is expressed as

$$Q = C\sqrt{\frac{2\Delta P}{\rho}}\frac{A_a}{\sqrt{\left(\frac{A_a}{A_b}\right)^2 - 1}},\tag{1}$$

where $C$ is the discharge coefficient which takes into account the viscosity of fluids, $\Delta P$ is the pressure difference across the venturi tube, $\rho$ is the density of the fluid, and $A_a$ and $A_b$ are the cross sections of the venturi tube at the two locations between which the pressure difference is determined. In the case of air, the density can be derived from the ideal gas law

$$\rho = \frac{PM}{RT},\tag{2}$$

where $P$ is pressure in Pascal, $T$ is temperature in Kelvin, $R$ is the gas constant and $M$ is the molar mass of air. Since the AIS measurement is based on the NTP assumption, the corrected sample flowrate ($Q_{\text{s,cor}}$) can be obtained by adding an additional multiplier, $\sqrt{\frac{T_{\text{atm}}}{P_{\text{atm}}}\frac{P_{\text{NTP}}}{T_{\text{NTP}}}}$, to Eq. 1 as follows

$$Q_{\text{s,cor}} = C\sqrt{2\Delta P}\sqrt{\frac{RT_{\text{NTP}}}{P_{\text{NTP}}M}}\frac{A_a}{\sqrt{\left(\frac{A_a}{A_b}\right)^2 - 1}}\sqrt{\frac{T_{\text{atm}}}{P_{\text{atm}}}\frac{P_{\text{NTP}}}{T_{\text{NTP}}}},\tag{3}$$

where $P_{\text{NTP}}$ and $T_{\text{NTP}}$ are pressure and temperature at the NTP condition, and $P_{\text{atm}}$ and $T_{\text{atm}}$ are at ambient atmospheric conditions. Eq. 3 can be simplified to the following form:

$$Q_{\text{s,cor}} = Q_{\text{s,meas}}\sqrt{\frac{T}{P}\frac{P_{\text{NTP}}}{T_{\text{NTP}}}},\tag{4}$$

where $Q_{\text{s,meas}}$ denotes the recorded sample flowrate by the AIS. Therefrom, the number concentration

of ions measured in each mobility range ($N_i$) is corrected by

$$N_{i,\text{cor}} = N_{i,\text{meas}}\cdot\frac{Q_{\text{s,meas}}}{Q_{\text{s,cor}}}.\tag{5}$$





In addition to the number concentration, the flowrates in the AIS influence the upper and lower limits of mobility ranges (Mirme et al., 2010; Mirme and Mirme, 2013). The lower and upper limiting mobility are proportional to the sheath flowrate ($Q_{sh}$) and the sum of sample and sheath flowrates, respectively.

Therefore, the corrected lower and upper limiting mobility can be expressed as

$$Z_{i,\mathrm{L,cor}} = Z_{i,\mathrm{L,meas}} \cdot \frac{Q_{sh,cor}}{Q_{sh,meas}} \ \text{ and } \ Z_{i,\mathrm{U,cor}} = Z_{i,\mathrm{U,meas}} \cdot \frac{Q_{sh,cor} + Q_{s,cor}}{Q_{sh,meas} + Q_{s,meas}}, \tag{6}$$

where $Z_{i,\mathrm{L}}$ and $Z_{i,\mathrm{U}}$ represent lower and upper limiting mobility in the mobility range, $i$.

In this work, to be comparable with particle data, air ion data are presented in Millikan mobility diameters. The conversion of electrical mobility to sizes is based on the Stokes-Millikan equation (e.g. Hinds, 1999), using measured ambient temperature and pressure. After the flow correction, the AIS has a measureable mobility size range of 0.9-48 nm.

The primary feature of an ambient AIS spectrum contains a persistent band of high ion concentrations at lowest sizes (Fig. 1), which is typically known as the cluster ion band. The upper boundary of this band typically lies at around 1.7 nm in the mobility diameter (~ 1.3 nm in the mass diameter (Mirme et al., 2007)) under mid-latitude ambient conditions, representing critical cluster sizes (Kulmala et al., 2013; Chen et al., 2016). After applying the flow and ambient condition correction to the AIS data, the cluster

ion size range at Dome C was found to be between 0.9 and 1.9 nm in the positive polarity (Fig. 1). However, the upper boundary of the cluster size range in the recorded negative AIS spectra was shifted down to around 1 nm, likely caused by the establishment of an ill-shaped electric field inside the analyser due to the malfunctioning of grounding or insulation. Such an anomaly is indicative of a low reliability of the negative ion data. Thus, only positive ion data from the AIS measurement is reported

in this work.





### 2.1.2.  Total aerosol particle measurement

A differential mobility particle sizer (DMPS), the same used by Järvinen et al. (2013), was responsible for recording the number size information of total aerosol particles. The DMPS classifies particles of mobility sizes in between 9 and 550 nm. It consists mainly of a medium-size Hauke-type DMA for

classifying particles and a TSI 3010 condensation particle counter (CPC) for detection. Prior to the classifying and detecting sections, the sampled air passes first through a bipolar radioactive charger (Ni-63), where aerosol particles in the sample attain a Boltzmann equilibrium charge distribution. These aerosol particles then enter the DMA. Particles of different electrical mobility are selected from the sample by changing the high voltage applied on the DMA inner electrode step-wisely. The

mobility-segregated particles are subsequently grown by vapour condensation and detected optically in the CPC. Each measurement cycle takes 10 min. The number size distribution of the measured aerosol particles was derived from the recorded mobility distribution via an inversion procedure described in detail by Aalto et al. (2001). The sizes of aerosol particles are presented in Millikan mobility diameters. Hereafter, unless otherwise specified, the particle and ion sizes refer to their Millikan mobility

diameters.

### 2.2.  Meteorological and LIDAR data

The ambient air temperature (T), relative humidity (RH), wind speed (WS) and wind direction (WD) data were from the routine meteorological observation at Station Concordia as part of the IPEV/PNRA Project - a collaborative project between "Programma Nazionale di Ricerche in Antartide" (PNRA) and

Institut Polaire Français Paul-Emile Victor (IPEV) (www.climantartide.it).

An automatic, depolarization LIDAR (532 nm) operates in Dome-C since 2008 (http://lidarmax.altervista.org/englidar/_Antarctic%20LIDAR.php). The LIDAR is sensitive to large aerosols and cloud particles, whose presence can be detected from a few meters above ground to 7000

m. The discrimination between liquid and solid relies on the aerosol-induced depolarization of linearly polarized laser light.





### 2.3. Derived quantities

To assist our analyses, growth rates, condensation sinks and formation rates of ions were determined from the measured air ion and total aerosol particle spectra. The growth rate characterises how rapidly particles enlarge in size due to condensational growth and coagulation, which typically has a unit of

nm/h (Dal Maso et al., 2005; Kulmala et al., 2012). The condensation sink describes the loss rate of condensable vapours onto aerosol particles and is expressed in $s^{-1}$ (Pirjola et al., 1998; Dal Maso et al., 2002). The formation rate of ions quantifies the rate at which ions in a certain size range are formed and has a unit of $cm^{-3}s^{-1}$ (Nieminen et al., 2011; Kulmala et al., 2012).

### 2.3.1. Growth rate (GR) determination

Air ion and total aerosol particle data are three-dimensional: the concentration of ions/aerosol particles evolves with both time and particle size. We used two approaches in determining GRs: the mode-fitting method (Dal Maso et al., 2005) and the appearance time method (Lehtipalo et al., 2014). The former approach follows the concentration change in the time dimension and the latter one follows the change in the size dimension.

In the mode-fitting method, at each time stamp of the measurement, a normal distribution is fitted to the measured concentration distribution along sizes in a logarithmic scale. The mode of the fitted curve is taken as the representative size of the bulk particles measured at this time stamp. In contrast, in the appearance time method, one determines the time stamp (the appearance time) at which the bulk

particles are considered to reach a certain size (the geometric mean size of a measurement size bin), based on the measured concentration distribution along time (Lehtipalo et al., 2014). This procedure is repeated for each measurement size bin. In this study, we defined the moment at which the concentration rises to 75% of the maximum as the appearance time for each size. The GR was then calculated as the slope of a linear fit to the size data as a function of time. In addition, we also

determined the instantaneous GR ($\mathrm{d}dp/\mathrm{d}t$) as the change in sizes within the interval of two adjacent time stamps to assist the analysis of the GR dependency on sizes.





### 2.3.2. Condensation sink (CS) determination

Sulphuric acid is considered as a key chemical species in forming aerosol particles in the ambient air (Kulmala et al., 2014). We determined CSs for sulphuric acid vapours from the number size distribution

measured by the DMPS, based on the method described by Pirjola et al. (1998) and Dal Maso et al. (2002), using the following equation

$$CS = 2\pi D \int_0^\infty d_p \beta(d_p) n(d_p)\, dd_p. \tag{7}$$

Here $D$ is the diffusion coefficient of condensing vapour molecules, $d_p$ is the diameter of aerosol particles, $n$ is the concentration, and $\beta$ is a transitional correction factor (Fuchs and Sutugin, 1971), which is a function of the mass accommodation coefficient ($\alpha$) and Knudsen number (Kn). In most applications, $\alpha$ is assumed to be unity while Kn can be connected to $D$ via the mean free path of vapour molecules ($\lambda_v$) using the mean free-path theory (Mason and McDaniel, 1988; Pirjola et al.,

1998).

The diffusion coefficient of vapour molecules is determined using the Fuller's model (Poling et al., 2004; Tang et al., 2014), which describes a binary gas system of species A diffusing in B:

$$D = \frac{0.00143 \cdot T^{1.75}}{[P/10^5 \text{:bar}] \cdot M_{AB}^{1/2} [(\Sigma_v)_A^{1/3} + (\Sigma_v)_B^{1/3}]^2}, \text{ with } M_{AB} = \frac{2}{1/M_A + 1/M_B}. \tag{8}$$

Here, $(\Sigma_v)_A$ and $(\Sigma_v)_B$ are the diffusion volumes of species A and B, respectively, and $M_A$ and $M_B$ are the corresponding molar masses. For the system of sulphuric acid diffusing in air, $\Sigma_{v,H_2SO_4} = 51.96$ and $\Sigma_{v,air} = 19.7$. The measured ambient temperature and pressure were used in the CS determination.





### 2.3.3. Formation rate of 2-nm ions ($J_2^\pm$)

The formation rates of 2-nm ions ($J_2^\pm$) was determined based on the following equation

$$J_2^\pm = \frac{dn_{2-3}^\pm}{dt} + \text{CoagS}_2 n_{2-3}^\pm + \frac{\text{GR}_{2\text{-}3}}{1\ \text{nm}} n_{2-3}^\pm + \alpha n_{2-3}^{\pm 2}, \qquad (9)$$

where $n_{2-3}^\pm$ is the ion concentration in the 2-3 nm size range, $\text{GR}_{2\text{-}3}$ is the growth rate of 2-3 nm ions and $\alpha$ is the ion-ion recombination coefficient ($\alpha = 1.6 \cdot 10^{-6}$ cm$^3$s$^{-1}$). $\text{CoagS}_2$ stands for the coagulation sink for 2 nm ions and it was determined using particle data measured by the DMPS based on the method described by Kulmala et al. (2001). Owning to the malfunction of the negative analyser in our

10 AIS, the ion formation rates in this study were determined only for the positive polarity, based on the assumption that the negative ion concentration equals to the positive ion concentration.

## 3. Results and discussions

### 3.1. Overview of features observed in air ions

During the campaign period, there were nearly 300 days with valid air ion measurements. A summary

15 of the event statistics of features seen in air ions at Dome C is presented in Table 1. We were able to identify new particle formation (NPF), wind-induced ion formation and cloud activation events from the valid measurements. Altogether, NPF events were observed on 32 days, wind induced ion formations on 36 days and cloud activations on 7 days with 2 definite cases. For the NPF events, 20 cases were classified as suppressed NPF events, which were characterised by no clear particle growth beyond 10

20 nm. Cases with two or more separate, yet simultaneous, NPF events with subsequent growth, i.e. multi-mode NPF events, were seen on 12 NPF days. Additionally, 85 days were recognised as event-free days, during which no above-mentioned events, contamination or other processes disturbing the cluster ion band (0.9 - 1.9 nm) were observed.





A clear seasonality was found on these event-free days in the cluster ion concentration (Fig. 2a). The cluster ion concentration was high during the warm months, with a maximum in February. The median cluster ion concentration was typically below 100 cm$^{-3}$ during the winter. This seasonality is related to variations in the natural ionising radiation, which produces initial charge carriers via the ionisation of air molecules, as well as to the availability of vapours capable of forming cluster ions (Chen et al., 2016). The variability in the ionising radiation originates, in general, from changes in the atmospheric radon concentration, terrestrial gamma and cosmic radiation (Chen et al., 2016). Owing to the presence of the snowpack, the contribution of radon exhalation or terrestrial gamma emissions from ground at Dome C to the ionisation of air molecules can be neglected. Also atmospheric radon emitted from the coastal areas or other remote regions may hardly reach Dome C by wind transportation. Therefore, the only major ionisation source of air molecules comes from cosmic radiation. The cosmic ray intensity is mainly modulated by the solar activity and therefore can be considered invariable in short terms. Muons are the main cosmic ray component responsible for the ionisation in the lower atmosphere (Goldhagen, 2000). The intensity of the muon flux in Antarctica has been found to have a weak seasonality and related to the stratospheric temperature, being low in winter and high in summer, and the temperature modulation was reported to be ± 8% on an annual basis (The IceCube Collaboration, 2011). Although the seasonal cycle in the cluster ion concentration corresponded to the seasonality in the muon flux, the influence of the variation in the ionising radiation alone is insufficient to account for the difference of more than 30 % in the cluster ion concentration between summer and winter (Fig. 2a). Thus, most probably the low winter cluster ion concentration was additionally affected by the limited source of vapours that can participate in cluster ion formation due to the inhibited photochemistry under the reign of polar-night darkness.

Markedly, even though NPF events were only seen in February, March, October and November, the median cluster ion concentration was always higher on NPF event days compared with event-free days. A similar phenomenon was also seen in a boreal forest environment at SMEAR II station in southern Finland (Fig. S1). Such a connection between cluster ion concentrations and NPF occurrences may imply that compared with event-free days, NPF event days probably had higher concentrations of





vapours that are able to contribute to both cluster ion formation and NPF. In the Antarctic autumn (February and March), the value of condensation sink (CS) tended to be higher on NPF days compared with event-free days (Fig. 2b). Since a large CS is indicative of a higher atmospheric sink for low volatility vapours, this observation suggests a particularly strong source rate of such vapours during this

time of the year, especially on NPF days. Interestingly, a reversed feature to the autumn pattern in CS was seen in the Antarctic spring (October and November, Fig. 2b).

## 3.2.  New particle formation (NPF) and growth

### 3.2.1.  NPF events

One of the major features observed in the AIS spectra was the process of NPF and subsequent growth of newly-formed charged clusters/particles. Seven clear NPF events were seen on consecutive days during 9-15 March, 2011, with the initial step traceable down to the cluster ion band (Fig. 3b). All these NPF events occurred during westerly winds, except the first one on 9 March (Fig. 3e). This NPF event was associated with winds from the contaminated sector (10-90°) and possibly affected by the diesel

generator of the station and motor vehicle pathways (Järvinen et al., 2013). These NPF events occurred after the sunrise between about 00:00 and 06:00 UTC (between 08:00 and 14:00 in the local UTC+8:00 time zone), which is in line with the proposed importance of photochemistry in NPF events and particle growths (Kulmala and Kerminen, 2008; Ehn et al., 2014; Kulmala et al., 2014). Except for the weak NPF event on 10 March, the newly-formed particles during 9-12 March reached sizes larger than 10 nm

and could be captured by both AIS and DMPS (Figs. 3a&b). Bumps in the concentration of 10-100 nm total particles corresponded to these NPF events (Fig. 3c). However, the NPF events seen during 13-15 March were restricted to sizes below 10 nm and showed no traces in the DMPS spectra (Figs. 3a&b). Consistently, no elevated concentrations were observed for particles in the size range of 10-100 nm for these events (Fig. 3c). Such differences result likely from the availability of vapours that sustain the

growth, which may be related to air masses coming from different origins. Slight concentration increases were perceptible in the cluster ion size range at the time when NPF events were initiated, but





no systemic features in relation to NPF events were identifiable in the total concentration of particles larger than 100 nm (Fig. 3c). The condensation sink (CS) varied similar to the total concentration of particles larger than 100 nm, ranging between $10^{-4}$ and $4 \cdot 10^{-4}$ s$^{-1}$, in consistence with the values of CS reported at Dome C during NPF events (Järvinen et al., 2013).

Figure 4 shows examples of multi-mode NPF events that were observed during 12-16 February 2011. One of them (14 to 15 February) had three concurrent NPF and growth events. The first of them was initially captured by the DMPS at around 03:00 UTC, and at a size of around 15 nm, but it showed no clear traces in the AIS measurements before 12:00 UTC. When the particles formed during this first

event were still growing in size, a second NPF event started from a size of 9 nm at 18:00 UTC and was detected almost simultaneously by both AIS and DMPS. Short after the onset of the second NPF event, and during the growth stages of both the first and second events, a third NPF event was observed in the AIS starting from the cluster size range. The growth of particles originating from this last NPF event ceased at around 6 nm and was therefore not seen by the DMPS. A similar multi-mode NPF event was

also observed on the following day, 16 February (Fig. 4). On 12 February at 06:00 UTC, a NPF event was observed by the DMPS at an initial size of 10 nm, and the same event was observed by the AIS after around 13:00 UTC. This event lasted until the noon of 13 February. Over the consecutive five days on 12-16 February, a slowly-growing population of 40-200 nm particles could be observed in the background, with their initial formation traceable back to 06:00 UTC on 12 February. Interestingly,

apart from the particles initiated at 10 nm and 40 nm, a third mode of particles with sizes larger than 100 nm was recognisable in the morning of 12 February. This particle mode underwent a slight growth during 12-13 February, and then gradually merged with the mode initiated at 40 nm at the end of 16 February. These multi-mode NPF events were associated with two times higher values of CS than the events presented in Fig. 3, owing to the presence of higher concentrations of background particles

(Figs. 3c and 4c&d).



### 3.2.2. Growth rate comparisons

The growth rates (GRs) determined by the appearance time method tended to be higher than those determined by the mode-fitting method (Fig. 5). This difference is likely to originate from the foundations that these two methods are laid on. Both methods were developed to treat the measured
number size distribution data of ions/particles that varies in time, size and ion/particle number concentrations. The appearance time method follows the concentration change as a function of time along the particle-size dimension, whereas the mode-fitting method tracks the concentration change as a function of particle size along the time dimension. Accordingly, the appearance time method is able to preserve the growth features related to particle sizes, whereas the mode-fitting method characterises
better the evolvement of particle growth with time.

The NPF event that occurred on 12 March between 00:00 and 6:00 UTC had clearly two simultaneously growing modes, with corresponding particle growth rates marked by $GR_6$ and $GR_7$ in Figs. 5a and b. For this NPF event, although higher uncertainties were associated with the GR determined by the
appearance time method than with the mode-fitting method, the former method succeeded in tracking the growth following both modes while the latter failed (Fig. 5). For $GR_1$ and $GR_4$, both methods led to similar GR values based on linear fitting, however, with a slightly smaller uncertainty (root-mean-square error) obtained using the appearance time method. Except for the slow growth cases, the appearance time method seems to present a narrower uncertainty range than the mode-fitting
method.

Even though the GR is often calculated as the slope of a linear fit to the size data as a function of time (Yli-Juuti et al., 2011; Lehtipalo et al., 2014), like in Fig. 5, it is not always appropriate to express the change in sizes along time by a linear proportionality, e.g. $GR_1$ determined by the appearance time
method in Fig. 5b and $GR_2$ determined by the mode-fitting method in Fig. 5a. Alternatively, the change in sizes within the interval of two adjacent time stamps, i.e. the instantaneous GR ($\mathrm{d}dp/\mathrm{d}t$), was used to investigate the size dependency of GR. We found that the instantaneous GR derived from the AIS





measurements during the NPF events tended to increase with an increasing particle size (Fig. 6c&d). A similar feature has been reported at many other sites for the sub-20 nm size range (see Häkkinen et al. (2013), and references therein). The median instantaneous GRs given by the appearance time method were in the range of 0.5 – 25 nm/h and by the mode-fitting method in the range of 1 - 150 nm/h. The

reported GRs of newly formed atmospheric aerosol particles are typically below a few tens nm/h (Yli-Juuti et al., 2011; Järvinen et al., 2013; Wang et al., 2017). The instantaneous GRs determined using the former method fall in this range, while using the latter method resulted in larger instantaneous GRs. This feature could be ascribed to the higher uncertainties associated with the mode-fitting method. The mode-fitting method tracks the mode concentration corresponding to sizes based on curve fitting

for each measurement cycle, and it could be that the sizes at which mode concentrations were identified apart significantly in two adjacent measurement cycles, i.e. over a short time interval. A large size difference over a small time interval, therefore, would lead to a huge instantaneous GR. In contrast, the appearance time method is based on looking for the time stamp, when the concentration reaches 75% of its maximum in the concentration vs. time space for each size channel of the instrument. Owing to the

fact that aerosol and ion data have a higher resolution in the time dimension than in the size dimension, the appearance time method could pick up the time stamp more precisely for each size than the mode-fitting method could do the sizes for each measurement cycle. Consequently, the appearance time method presents GRs with smaller uncertainties (Fig. 5) and yields more representative instantaneous GRs. The instantaneous GRs derived from the DMPS measurement exhibit a rather vague pattern in

connection to sizes (Figs. 6a&b); yet, a light increasing tendency might be still deducible.

### 3.2.3. Formation rates of 2-nm positive ions ($J_2^+$)

The formation rate of 2-nm positive ions ($J_2^+$) was determined for 26 NPF event days, for which GRs in the 2-3 nm size range were obtained using the appearance time method. The average value of $J_2^+$ was

0.014 cm$^{-3}$ s$^{-1}$, with a standard deviation of 0.020 cm$^{-3}$s$^{-1}$. Other characteristic values for $J_2^+$ were the followings: 0.0005 cm$^{-3}$s$^{-1}$ (minimum), 0.0024 cm$^{-3}$s$^{-1}$ (1$^{st}$ quartile), 0.0066 cm$^{-3}$s$^{-1}$ (median), 0.015





cm$^{-3}$s$^{-1}$ (3$^{rd}$ quartile), and 0.079 cm$^{-3}$s$^{-1}$ (maximum). These ion formation rates are comparable to those reported for the SMEAR II station in Finland (Nieminen et al., 2011), as well as to those observed in several other sites in Europe (Manninen et al., 2010).

### 3.3. Other specific features

#### 3.3.1. Influence of cloud/fog formation on aerosol particles and air ions

A cloud activation event initiated at around 14:00 UTC was observed on 20 January 2011 (Fig. 7). In general, such events are characterised by a disappearance of aerosol particles from the measured particle size range (Komppula et al., 2005; Kyrö et al., 2013). Additionally, a sudden drop in the cluster ion concentration has been reported as a feature for cloud activation events (Lihavainen et al., 2007). We observed similar connections between cluster ions and cloud activation at Dome C (Figs. 7a, b&c). Moreover, we found that the cloud activation event was accompanied by a burst of ions in the 8-42 nm size range measured by the AIS (Figs. 7b&c), yet not captured by the DMPS.

By following the approach introduced by Komppula et al. (2005) based on DMPS measurements, it can be estimated that particles larger than about 110 nm in diameter had been activated into cloud droplets during this cloud event (Fig. 7). As a consequence, cluster ions were lost efficiently onto the cloud droplets. Because of their large sizes beyond the detection capability of the DMPS, cloud droplets could collect multiple charges via the uptake of cluster ions, which might then be detected by the AIS as singly charged particles in the size range of 8-42 nm. Additionally, these ions detected by the AIS might also be artificial products resulted from the cloud droplet cleavage inside the sampling line of the instrument, possibly related to the high sample flowrate and the low temperature condition (Fig. 7d). This cloud activation was possibly a result of ground-level fog formation, as LIDAR observations showed no evidence of clouds above 40 m (Figs. S2a&b). A cloudy pattern appeared in the LIDAR spectra at the near-ground level after about 15:00 UTC (Figs. S2c&d). This one-hour delay compared with the aerosol instruments may be related to the fact that aerosol particles/cloud droplets cannot be captured by the LIDAR unless the aerosol/cloud layer is optically thick enough. It is very likely that the




ground-level fog was initially very thin, yet observable with the DMPS system, and only later became thick enough for the LIDAR to capture it. The high depolarisation indicates a high probability of the presence of ice particles. Some precipitation could be recognised (Figs. S2a&b), originating from cirrus clouds at heights between 2500 and 3000 m above ground between 16:00 and 18:00 UTC. Light

precipitation might have reached the ground level after around 18:00 UTC (Figs. S2c&d), which perturbed the cloud activation and impaired the effect of fog/cloud formation exerted on air ions (Figs. 7a-c).

### 3.3.2. Wind-induced ion formation

Ion formation events during strong wind episodes have been observed at Aboa in Antarctica (Virkkula

et al., 2007), as well as at the high-altitude site on Jungfraujoch in Switzerland (Manninen et al., 2010). At Dome C, we observed wind-induced ion formation especially during the winter months. An example of such an event, observed during 3-4 July 2011, illustrates the close connection between the ion formation and wind speed (Fig. 8): ions generated by a strong wind were mainly in the cluster ion size range, even though a large number of ions were also apparent in the 1.9-10 nm size range.

Under strong wind conditions, small snowflakes and ice crystals in the surface layer of the accumulated snow on the ground can be resuspended by turbulence and be shattered further by their collisions (Pomeroy and Jones, 1996). Vapours adsorbed on and trapped in these snowflakes and ice crystals can be released into the air to replenish vapours in the air that are capable of participating in cluster ion

formation and possibly also NPF. In addition, this resuspension process also assists the escape of vapours trapped beneath the surface snow layer on the ground. Moreover, owing to the sudden drop of the surrounding vapour pressure, gaseous molecules of water and other trace species may also be freed from the resuspended particles by sublimation (Pomeroy and Jones, 1996). Ionising radiation produces primary ions via ionisation, which are either lost through ion-ion recombination or transformed into

more stable air ions by nucleation or condensation (Chen et al., 2016). A small concentration of ions slightly larger than the cluster sizes could be observed in connection to the high wind speeds between




6:00 and 12:00 UTC (Figs. 8b&c). As the wind speed increased further after 12:00 UTC, the vapour replenishment was probably amplified, leading to an ion burst in the size range of 0.9-10 nm via nucleation, condensation and coagulation. A fraction of these ions seems to be able to further undergo dynamic processes to form large aerosol particles with sizes of even 500 nm (Fig. 8 a).

Turbulent conditions may enhance the collection of electric charges by the shattered snowflakes and ice particles via charge transfer from initial charge carriers, contributing to the formation of an ion burst. In addition, the shattered particles may gain electric charges through friction charging. However, these two pathways of ion formation may not be of a high likelihood to contribute to the ion burst captured by the

AIS. In principle, shattering of resuspended snowflakes and ice particles mechanically by turbulence results in the formation of particles of smaller but random sizes. If this mechanism had produced nano-sized particles that subsequently became electrically charged either by charge transfer or friction charging, our AIS should have detected some of them and have shown an unsystematic spectrum, i.e. ions of random sizes and concentrations. Yet, on the contrary, the AIS showed high concentrations of

ions of only small sizes, and hardly anything of sizes larger than 2-3 nm in diameter between 12:00 and 14:00 UTC during the intensification of the wind speed. Nevertheless, the involvement of these processes may not be ruled out completely based on our ambient observations. This kind of wind-induced ion formation would be worth further experimental investigations, for example by releasing snow in a wind tunnel to disclose the true mechanisms governing the ion production.

By putting together all 36 wind-induced ion formation events, a linear correlation was identified between the logarithm of the ion concentration and wind speed (Fig. 9), similar to the feature reported at Aboa (Virkkula et al., 2007). However, the effect of wind on ions seemed to be stronger at Dome C than at Aboa (Fig. S3). The slopes for the logarithm of ion concentrations as a function of wind speeds

differed by about an order of magnitude between Dome C and Aboa. This is so far the clearest and largest difference in the air ion processes at these two sites and deserves a more detailed study in the future.





## 4. Conclusions

Based on one year of air ion observations with an air ion spectrometer (AIS) at Dome C, Antarctica, we found that this site has an unexpectedly rich set of ion processes, especially when considering its inland location on the largest ice desert on the Earth - the Antarctic plateau. New particle formation (NPF),

wind-induced ion formation and ion production and loss associated with cloud/fog formation were the main processes that were found to modify the number size distribution of air ions at this high-altitude site. On event-free days, i.e. on days without the above-mentioned processes or other anomalies, concentrations of cluster ions (0.9-1.9 nm) showed a clear seasonality, with high concentrations in the warm months and low concentrations in the cold season. Days with NPF events were characterized by

higher cluster ion concentrations than event-free days. The specific features of the recorded air ion data allowed further classification of NPF into suppressed NPF and multi-mode NPF events. The former refers to NPF events, during which the growth of newly-formed particles hardly exceeds 10 nm, and the latter characterises NPF events with two or three co-occurring NPF and growth events in different size ranges.

Growth rates (GRs) determined using the mode-fitting method and appearance time method were used to characterise the NPF processes. The comparison between these two methods suggest that the GRs derived from the appearance time method work better in depicting the cases with a fast particle growth, whereas GRs determined from the mode-fitting method appeared to be more suitable for describing

cases with a slow particle growth. We found that the change in particle diameters did not usually increase linearly with the time. Therefore, we derived the instantaneous GR ($ddp/dt$) as the change in sizes within the interval of two adjacent time stamps, and found that the GR tended to increase with an increasing particle size. The formation rate of 2-nm positive ions was found to be $0.014\pm0.020$ cm$^{-3}$ s$^{-1}$ based on 26 NPF events.

Ion production in relation to cloud/fog formation in the size range of 8-42 nm was found unique at Dome C. These ions may be either multiply-charged particles detected as singly charged in the AIS, or



splinters of cloud droplets formed inside the instrument related to the instrumental behaviour under the extremely cold conditions. Accordingly, it would be worthwhile to carefully characterise the instrumental behaviour of ion spectrometers under extremely low temperature conditions at the presence of cloud droplets by conducting laboratory experiments. In addition, wind-induced ion formation was found to resemble new aerosol particle formation from vapours released from the snow, rather than being caused by mechanical charging of shattered snowflakes or ice crystals. The ion formation during strong wind episodes is a phenomenon of great interest. It is also worth mentioning that at this high-plateau site, wind-induced ion formation was approximately an order of magnitude stronger than at the low-altitude Antarctic site, Aboa, in which the same phenomenon has been observed earlier. The hidden mechanisms behind such processes need further investigation, since it may present a new pathway of atmospheric NPF in dark wintertime conditions.

The air ion data used in this work was limited to the positive polarity due to a technical malfunctioning of the negative analyser. Further ambient measurements on air ions would be valuable to be carried out at Dome C and other sites on the Antarctica plateau, not only to reveal possible differences between positive and negative ion properties and their connections to the ion and aerosol processes, but also to acquire a better characterisation on atmospheric NPF in Antarctica and to understand the mechanisms behind the ion formation related to the cloud/fog formation or wind episodes.

Acknowledgement

This work received financial support from the Academy of Finland (projects no. 264375 and 264390), the NordForsk funded Nordic Centre of Excellence CRAICC (Cryosphere-atmosphere interactions in a changing Arctic climate, projects no. 26060), and the Academy of Finland's Centre of Excellence program (Centre of Excellence in Atmospheric Science – From Molecular and Biological processes to the Global Climate, project no. 272041). Funding for this research was also provided by Consiglio Nazionale delle Ricerche and PNRA (projects 2009/B.04 and 2010/A3.05). We appreciate the support of the IPEV/PNRA Project "Routine Meteorological Observation at Station Concordia",



<www.climantartide.it> with the radiosounding data set. Xuemeng Chen acknowledges the Doctoral Programme in Atmospheric Sciences (ATM-DP, University of Helsinki) for financial support. Valuable advice from Dr. Sander Mirme is sincerely appreciated.

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

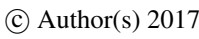



Figures & Tables

Table 1. Summary of different types of features in the observed air ions at Dome-C. Suppressed new particle formation (NPF) indicates that the extent of the subsequent growth after NPF hardly exceeds
5   10 nm. Multi-mode NPF characterises events of two or more separate NPF and subsequent growths. The numbers of suppressed NPF and multi-mode NPF days are subsets of the total number of NPF days. The number of possible cloud activation cases is shown in brackets. Event-free days include all the days, on which no NPF, wind-induced ion formation, cloud activation, contamination due to local maintenance activities, or other unexplainable processes disturbing the cluster ion band.

| Measurement length (days) | Valid ion measurement (days) | Event-free (days) | NPF (days) | Suppressed NPF (days) | Multi-mode NPF (days) | Wind-induced ion formation (days) | Cloud activation (days) |
|---|---|---|---|---|---|---|---|
| 330 | 287 | 85 | 32 | 20 | 12 | 36 | 2 + (5) |





Table 2. Growth rates estimated from linear fittings with root-mean-square errors expressed as uncertainties for the nine growth modes shown in Fig. 5. $GR_{mf}$ stands for growth rates determined by the mode-fitting method and $GR_{apt}$ for those determined by the appearance time method. Blue text in the table corresponds to GRs determined from the AIS measurement (9 – 12 March, 2011) and black text to
5   GRs determined from the DMPS measurement (25 Feb., 2011).

| ID | $GR_1$ | $GR_2$ | $GR_3$ | $GR_4$ | $GR_5$ | $GR_6$ | $GR_7$ | $GR_8$ | $GR_9$ |
|---|---|---|---|---|---|---|---|---|---|
| **$GR_{mf}$ [nm/h]** | **1.4±2.2** | **0.4±0.7** | **0.3±0.3** | **0.3±0.3** | **0.5±1.0** | **0.9±0.3** | **0.9±0.3** | **0.5±1.0** | **1.2±0.7** |
| Size ranges [nm] | 4.6-29 | 1.5-10 | 6.7-14 | 1.4-4.7 | 1.3-12 | 1.3-8.6 | 3.1-8.2 | 1.7-12 | 10-17 |
| **$GR_{apt}$ [nm/h]** | **1.3±1.2** | **1.1±0.5** | **1.1±0.8** | **0.4±0.2** | **1.1±0.5** | **1.5±1.2** | **3.5±1.0** | **1.6±7.5** | **3.3±1.2** |
| Size ranges [nm] | 5.3-31 | 1.4-17 | 7.1-17 | 1.4-5.3 | 1.3-20 | 1.4-13 | 1.3-13 | 1.7-17 | 11-20 |





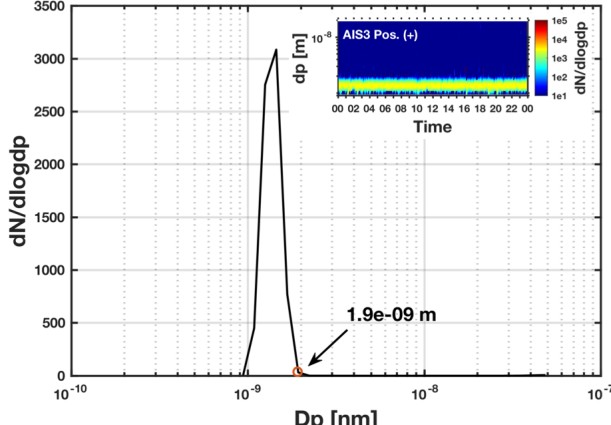

Figure 1. The median size distribution of ions measured by the AIS on an event-free day (16 January, 2011). The measured number size distribution of this day is shown in the contour plot.



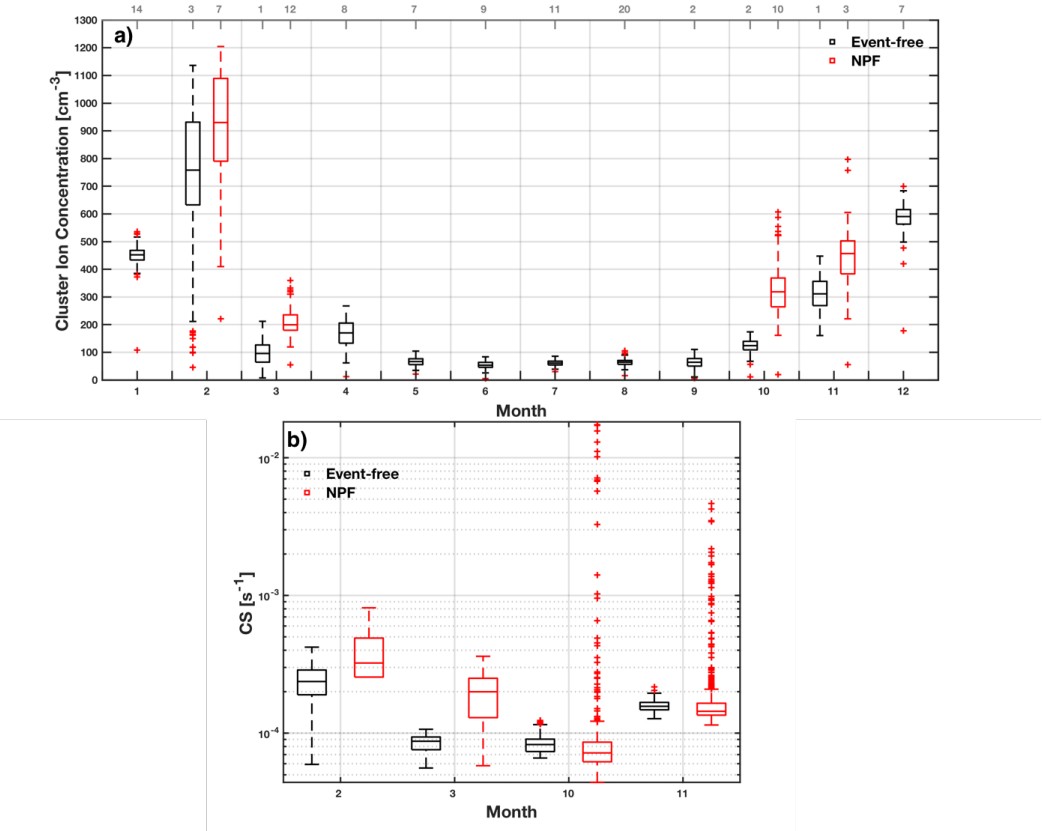

Figure 2. Seasonality in the median cluster ion (0.9 - 1.9 nm) concentration. Tops and bottoms of the boxes are the 75th and 25th percentiles of the median daily cluster ion concentrations in 10 min time resolution, with bars in the middle showing the 50th percentiles. Whiskers represent spans of the interquartile ranges multiplied by 1.5. Cluster ion concentrations on new particle formation (NPF) days shown in red and on event-free days in black. Event-free conditions were restricted to days, on which no NPF, cloud activation, wind-induced events or contamination as well as other anomalies altering the ion concentration in the cluster band. The numbers of days classified as either event-free or NPF are displayed on the top of the panel in grey colour.





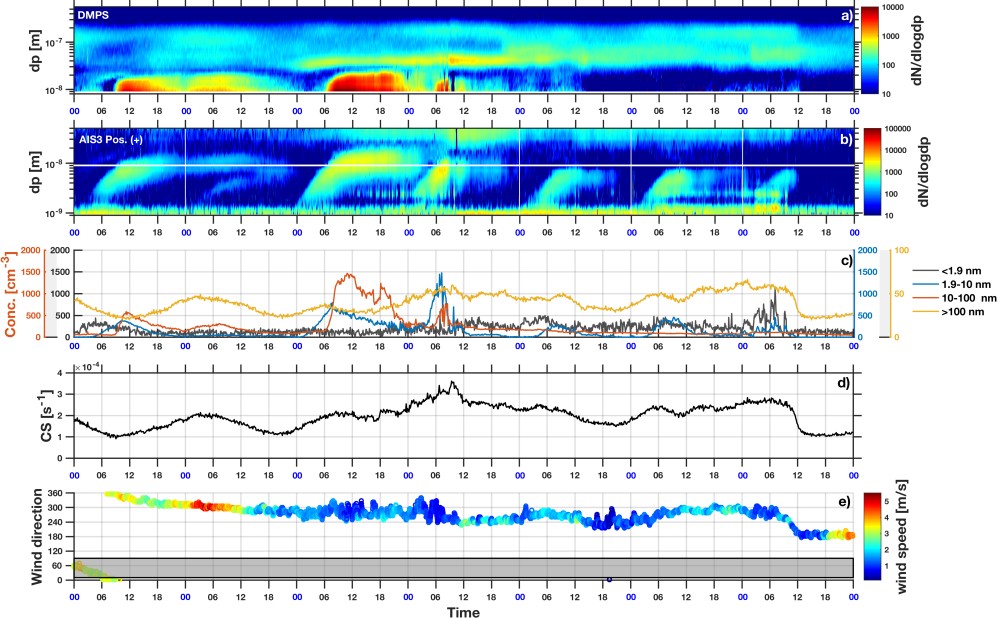

Figure 3. Consecutive new particle formation events observed during 9-15 March 2011. a) DMPS spectra; b) AIS positive polarity spectra; c) ion and particle concentrations in four different size ranges: 0.9-1.9 nm (ions), 1.9 - 10 nm (ions), 10 - 100 nm (total particles), and total particles of diameters above 100 nm; d) CS; and e) wind direction, colour-coded with wind speed. The white line in b) indicates the lower limit of the DMPS size range. The grey band in e) represents the contaminated wind sector. Data are presented in UTC.





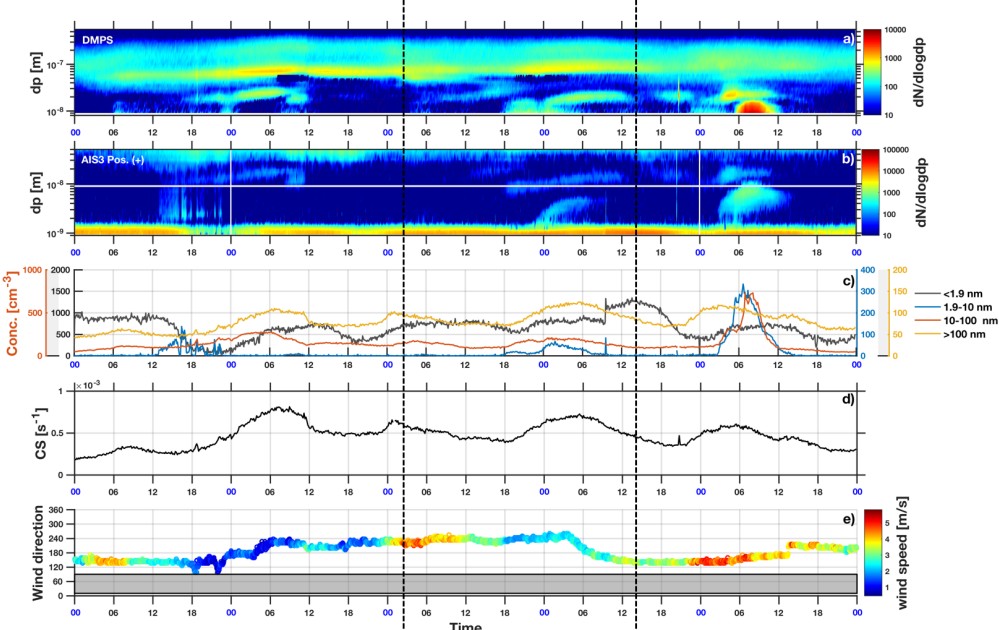

Figure 4. Consecutive multi-mode formation and growth events observed during 12-16 February 2011. a) DMPS spectra; b) AIS positive polarity spectra; c) ion and particle concentrations in four different size ranges0.9-1.9 nm (ions), 1.9 - 10 nm (ions), 10 - 100 nm (total particles), and total particles of diameters above 100 nm; d) CS; and e) wind direction, colour-coded with wind speed. The white line in b) indicates the lower limit of the DMPS size range. The grey band in e) represents the contaminated wind sector. The two vertical dashed lines are to outline the three NPF occurred on 14-15 February 2011. Data are presented in UTC.



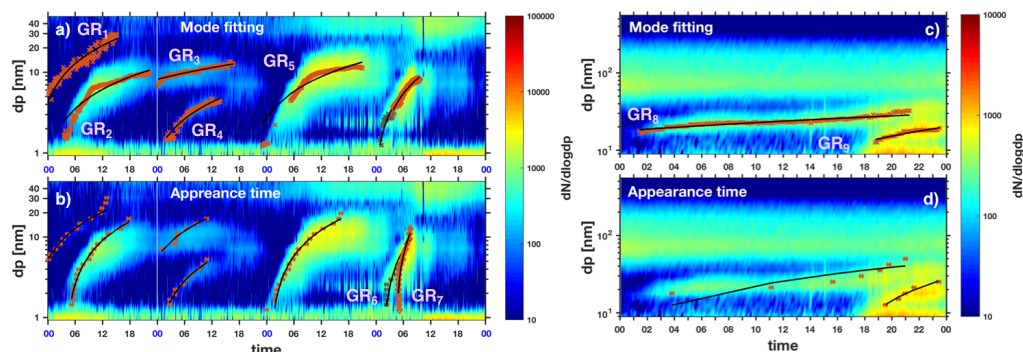

Figure 5. Comparison of growth rates for nine growth modes (GR$_i$, $i$ = 1…9 in figrue) determined by the mode-fitting method (a & c) and by the appearance time method (b & d), from the AIS measurement for 9 – 12 March 2011 (left-hand-side column) and from the DMPS measurement for 25 Feb. 2011 (right-hand-side column). Growth rates estimated from linear fittings with root-mean-square errors expressed as uncertainties are shown in Table 2. The red or grey dots depict the estimated size evolvement of ions/particles with respect to time determined by the mode fitting or appearance time methods, with the linear fits to these size-time relationships shown as black lines.



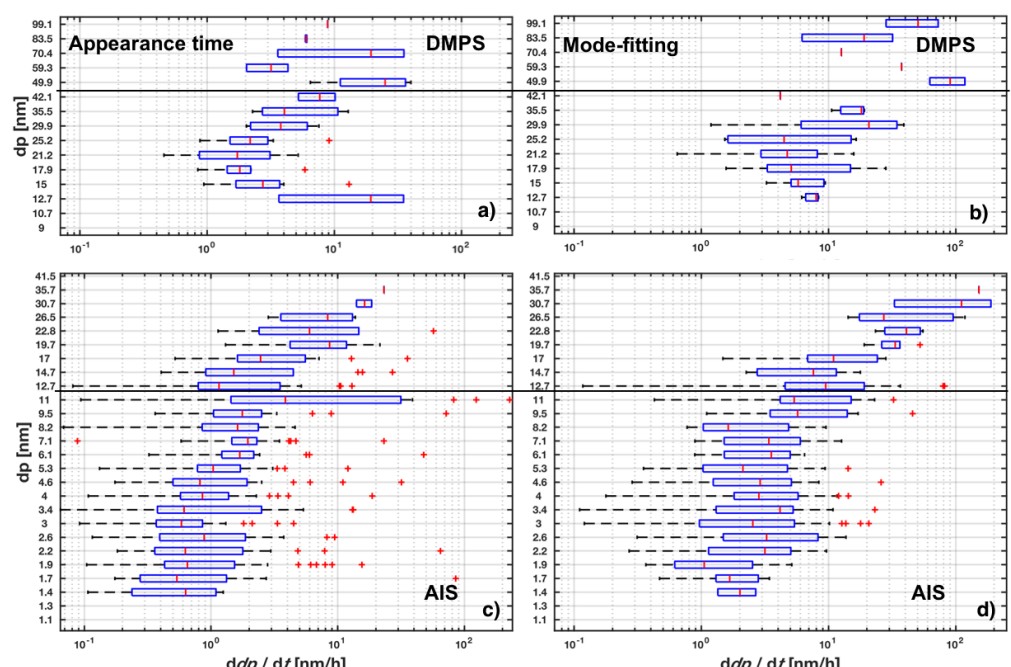

Figure 6. The growth rate dependency on size. Sizes are in mobility diameters and growth rates of ions and particles are presented as discrete time derivatives of the change in mobility diameters ($\mathrm{d}dp/\mathrm{d}t$). Growth rates of aerosol particles measured by the DMPS shown in the upper panel, a) & b), and growth rates of ions measured by the AIS shown in the lower panel, c) & d). Growth rates in the left-hand-side column, a) & c), are determined using the appearance time method and those in the right-hand-side column, b) & d), using the mod fitting method. The solid black lines indicate the overlapping size range of the DMPS and AIS measurements. The box was drawn with 25th and 75th percentiles of GRs determined at each size, with the median indicated as a right bar inside the box. The whiskers extended to the smallest and highest GR values within a 1.5 times the interquartile range at each size. GRs beyond the 1.5 times interquartile range were marked by red crosses as outliers.





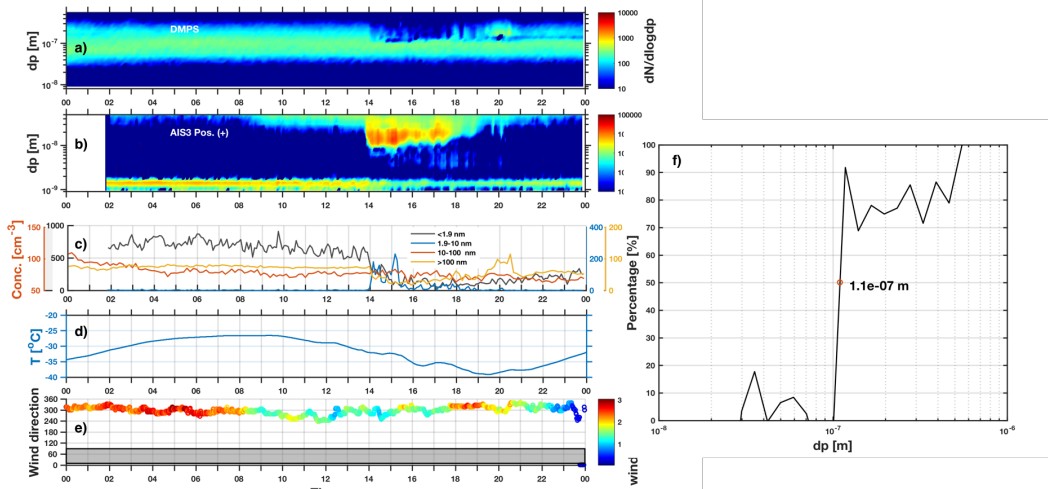

Figure 7. A cloud activation event observed on 20 January 2011. a) DMPS spectra; b) AIS positive polarity spectra; c) ion and particle concentrations in four different size ranges: ions of diameters below 1.9 nm, ions of diameters between 1.9 and 10 nm, total particles of diameters between 10 and 100 nm, and total particles of diameters above 100 nm; d) Ambient air temperature ($T$); e) wind direction, colour-coded with wind speed; and f) activation diameter ($D_{50}$), determined based on the method described by Komppula et al. (2005). Data are presented in UTC.



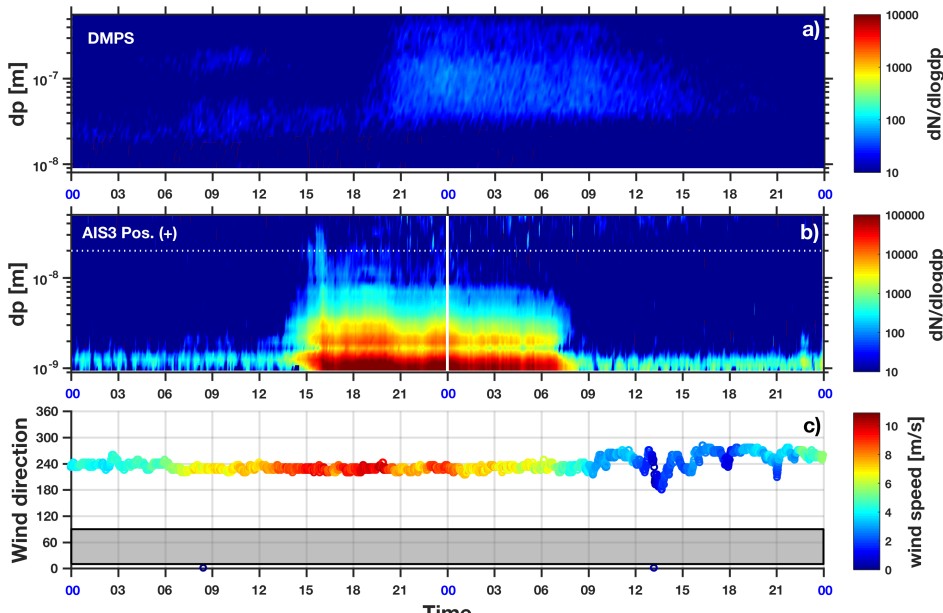

Figure 8. A wind-induced ion formation event observed on 3-4 July 2011. a) DMPS spectra; b) AIS positive polarity spectra; and c) wind direction, colour-coded with wind speed. The white dotted line is to outline the location of 20 nm for eye guiding. The grey band in c) represents the contaminated wind sector. Data are presented in UTC.





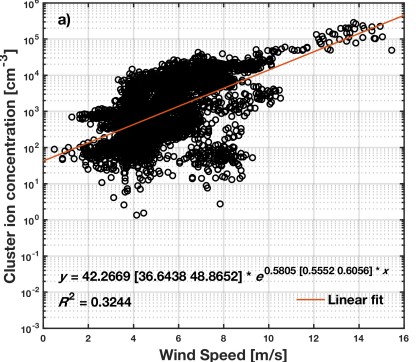 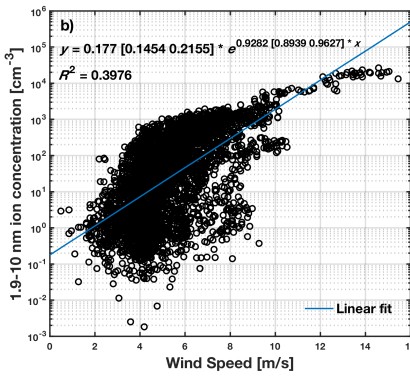

Figure 9. Ion concentrations as a function of wind speeds. a) ion concentration in the cluster size range (0.9 - 1.9 nm) and b) ion concentration in the size range of 1.9 and 10 nm. The solid lines are linear fits to the data with 95% confidence bounds of the coefficients shown in the brackets. $R^2$ is the coefficient of determination measuring the goodness of fit, which denotes the fraction of the total variation in the data can be explained by the fit.