# Peer review of "Features in air ions measured by an Air Ion Spectrometer (AIS) at Dome C"

_Atmospheric Chemistry and Physics, 2017_

## Referee Comment (RC1) · Anonymous Referee #1 · 12 May 2017

The article at hand reports on virtually year-round air ion size distribution spectra measured for the first time in continental Antarctica (Dome C). An in-depth evaluation of the comprehensive AIS record along with simultaneous differential mobility analyser (DMA) as well as LIDAR measurements revealed several highly interesting and novel findings considering (i) seasonality of air ion occurrence, (ii) new particle formation (NPF), (iii) size dependent growth rates, wind induced ion formation and (iv) the impact of cloud/fog formation on air ion production. In my opinion, this clearly written manuscript presents invaluable results to elucidate aerosol formation processes above continental Antarctica in general and in particular with respect to ion induced NPF. Thus, the manuscript is certainly appropriate to ACP and I recommend a publication after few rather minor revisions specified below.

Specific comments: Abstract (page 2, lines 11-13): Odd sentence, consider revision.

Introduction (page 4). For non-experts, I especially miss a short statement about cosmic ray intensity at Dome C compared to mid-latitudes. In addition: Is the total intensity of ionising radiation comparable to continental mid-latitudes?

Methods (page 4, lines 16-19): I wonder if the experiments were installed in a separate hut somewhat upwind to the main station as described in Järvinen et al. (2013) or Becagli et al. (2012). In this regard, the authors should briefly address the potential problem of local contamination.

Chapter 3.2.1 (page 13): Is it possible to assess the impact of neutral clusters on NPF? Definitely, this may be an additional important issue for dedicated future investigations at this site (see Conclusions, page 21, last section).

Chapter 3.2.2 and 3.2.3: Evaluation of particle growth rates and ion formation rates as described presupposes that NPF occurred in homogeneous air masses, thus neglecting the potential role transport and mixing processes. I think it is worthwhile to allude to (and discuss) the impact of these processes on the variance of particle growth- and ion formation rates, especially in regard with the derived extraordinarily high instantaneous growth rates up to some 100 nm/h.

Chapter 3.3.2 and Figure 9: In the Introduction, the particular conditions at Dome C, i.e. pronounced ionisation rates but limited source of vapours for clustering, were stressed (page 4, lines 3-7). Looking at Fig. 9a, I realized that ion concentrations in the size range 0.9 – 1.9 nm were roughly about a few hundred at low wind speeds (< 5m/s), which is in turn comparable to ion concentrations observed in a boreal forest (Chen et al., 2016; Figure 9 therein, sum over size range 0.8 – 1.7 nm), a site where sources of condensable vapours should not be limited. Does this analogy indicate that higher ionisation counterbalanced the lack of condensable vapours at Dome C? I think it may be worthwhile to speculate about this point.

Table 1: This table is redundant and can readily be removed, because it provides no further information as already presented in the main text (Chapter 3.1, page 11). If

at all, a plot showing the occurrence of the different features during the observation period on a time scale could be much more enlightening.

---

## Referee Comment (RC3) · Anonymous Referee #3 · 23 Aug 2017

The article presents an analysis of data of air ions (size range 0.9 to 42 nm) taken between the 22nd of December 2010 and 16 November 2011 at the Dome C station in Antarctica. The article focuses on the first steps of new particle formation providing formation rates and growth rates. The Authors give an overview of the main features of the size distribution of the ions relating them with measured meteorological parameters, such as ion formation/loss induced by wind and by the interaction with cloud/fog droplets.

In my opinion, the manuscript is well written in most of its parts and presents data that are very interesting for the atmospheric aerosol community. However, the Referee thinks that the manuscript is punctuated with generic statements and lack of specific numbers/information. The clarity and the relevance of the paper would be increased

by fixing those issues. Furthermore, the Referee feels that some statements in paragraph 3.1 and 3.3.2 are highly speculative and need a more thorough explanation and references to substantiate the statements. For these reason the Referee recommends publication after a revision that involves the rewriting of these parts.

Specific comments:

Major comments:

I) Page 9 line 16 to 21: The Referee is familiar with both methods to calculate the growth rates (mode fitting and appearance time) and still had hard time to follow the Authors' explanation. Please rephrase in a way that can be obvious also to the non-experts. Consider to add a figure in the supplementary material if the wording doesn't get any easier.

II) Page 12 line 8 to 10: "Owing ... the ionisation of air molecules can be neglected". The Referee thinks that this statement is highly speculative and needs to be modified. Are there measurements in the literature that support this claim? If so, please cite them. Furthermore, even if there is a snow pack, radon could make its way through the snow crystals and gamma radiation would need at least several tens of centimeter of solid ice to be shielded effectively. In addition, there can be areas, such as vertical walls that can be snow free and made of rock that is rich in uranium. If this is not the case, please state it in the paragraph with, if possible some references. Otherwise, rephrase the statement signaling its speculative nature.

III) Page 13 line 1 to 6: As CS tended to be higher during NPF can author exclude that NPF was generated by condensable vapors from "polluted" air masses from the continent? The Referee thinks that a look at back trajectories is needed in order to understand weather the source of vapors is local or from long range transport. This information would be very valuable and increase the relevance of the paper.

IV) Page 13 line 24 and 25: The Referee thinks that a look at back trajectories might

help the Authors to be less speculative about the origin of the air mass. Therefore, the Referee feels strongly that a back trajectory analysis should be added and discussed.

V) Chapter 3.2.2: A discussion on the uncertainties of the GR is completely lacking. The Referee thinks that must be added and discussed in relation to the variability of the GRs calculated with the 2 different methods.

VI) Page 16 line 23 to 26: The Referee thinks that an estimate of the uncertainty of J2+ is necessary in order to better assess the significance of the statistical parameters included in the text. In other words, is the standard deviation of the J2+ comparable, larger or smaller with respect to the uncertainties? Please add the missing information and discuss.

Minor comments:

Abstract

1) Page1, line 24: "One ... days." Rephrase. It is unclear if the Authors want to refer to event days or event free days.

2) Page 2 Line 4 and 5: The Authors use 2 time the expression "work better" which is very vague. Please rephrase using a more precise terminology.

3) Page 2 Line 7 and 8: "The ion production ... seemed a unique feature at Dome C ..." Please be more specific. A unique feature in Dome C relative to the whole world? Relative to the other data collected in Antarctica?

4) Page 2 Line 10: "cleavage" replace with a word/rephrase using wording that is understandable to a wide audience.

3) Page 2 Line 11 and 12 "our observation ... ice crystals." This is a very interesting result! The Referee thinks that it should be explained more thoroughly in the appropriate paragraph (see comment XX)

Introduction 4) Page 2 line 17: Please add the size/mobility ranges next to "primary

ions" and "aerosol particles" for a more complete information for the reader that might be new to atmospheric ions.

5) Page 3 line 2:"Such charged nanoparticles ... are typically observed during new particle formation (NPF) events." Confusing. Stable air ions are observed also without NPF. Please rephrase.

6) Page 3 line 9 to 20: "Carslaw et al. (2013) ... years (Fiebig et al., 2014)." Confusing paragraph. Please add a closing sentence that explains why this work is important and how it is different from the cited previous work in this way the list of previous papers on the topic will make more sense to the reader.

Methods 7) Page 5 line 4 and 5. Please add the flow rate after "high sample flow rate", which is too vague to be useful. The Referee is aware that there is a flow rate some 20 lines below, but the use of "High" is not helpful and repetita iuvant.

8) Page 7 line 21 to 23: The Referee thinks that a plot of the shifted spectrum added to the supplementary material could be very useful to the readers and AIS/NAIS users. I recommend adding it.

9) Page 9 line 4: "growth and coagulation". Remove coagulation.

10) Page 9 line 10: "Air ion and total aerosol particle data are three dimensional: " Remove. It is confusing and unnecessary.

11) Page 9 line 12 and 13: the Authors refer to " mode fitting method" and "appearance time method" throughout as "former" and "latter". Please spell them out each time. Readers will have easier times at understanding to which method the Authors are referring to.

12) Page 9 line 26: "assist". Confusing term. Please rephrase.

13) Page 3 line 3: "Sulphuric acid is considered as a key chemical species". remove "as".

14) Page 11 line 14:"During the campaign period, there were nearly 300 days with valid air ion measurements". Please add what is the corresponding percentage of valid days as well. The Referee is aware that the Authors refer to Table 1 where this information can be retrieve, but thinks that would be of help to the reader to make it explicit in the text.

15) Page 11 line 18: "definite". Confusing adjective. Please rephrase.

16) Page 12 Line 2: "high" please add a value e.g., median or a range of warm months to help the reader.

17) Page 12 line 4: "natural ionising radiation" is a confusing term as cosmic rays are also natural and ionizing. Please change wording.

18) Page 12 line 24: remove "Markedly" it is unnecessary. Please reword the sentence to make it more understandable/intuitive. In addition, do the Authors have information about the boundary layer (BL) in February, March, October and November? Can the Authors exclude that some of those high cluster ion concentration are due to limited mixing due to shallow BL? Please elaborate and include in the text.

19) Page 13 Line 20: "Bumps ... NPF events (Fig. 3c)". Please rephrase being clearer. Consider replacing the word "bumps" with e.g., "sudden increase".

20) Page 13 line 24 and 25: "Such differences result ... different origins". Replace "differences" with "different".

21) Page 13 line 26: replace "perceptible" with "measurable"

22) Page 14 line 11: "Short after". Please be less vague, remove "short" and add temporal information in the text.

23) Page 14 line 18 to 21: "slowly growing" ... "slight growth". Please add next to this general expression the value of the GR. It will make the text more useful to the reader.

24) Page 14 line 24: Replace "owing to" with "because of"

25) Page 16 line 19: "... and yields more representative instantaneous GRs". The word "representative" should be reserved to statistical analysis. If such analysis was done to assess whether the calculated GRs were statistically representative please add a sentence about the method used, otherwise rephrase.

26) Page 16 line 20: The Referee thinks that a comparison between the GR for ions with Dp>10 nm and particles in the same size range should be discussed in this paragraph.

27) Page 17 line 17: "...  110 nm". Based on the data analyzed in this work and on literature is the activation in this size range typical? The Referee recommends, if possible, adding some discussion about this in this paragraph.

28) Page 18 line 11: "we observed wind-induced ion formation especially during the winter months". Please add some numbers to make this statement less vague. How many times in winter with respect to other months?

29) Page 18 line 24: " Ionising radiation produces primary ions via ionisation". Please either remove "via ionization" or add what is ionized e.g., "via ionization of vapor molecules"

30) Page 19 line 8: "However ...  contribute to the ion burst captured by the AIS". The Referee thinks that this statement is highly speculative and unsubstantiated. The Authors should discuss more in length adding references.

31) Page 19 line 22: "similar" the wording makes the sentence too generic and vague, please discuss further, how those feature are similar.

32) Page 20 line 3: "unexpectedly" unnecessary adjective, please remove.

33) All figures of DMPS and AIS size distributions: add units to dN/dlogDp make the units of Dp consistent (all nm or all m) add thick labels so that are consistent and at least 2 in number e.g., 10 and 100 nm

34) Page 31 caption of figure 1: specify the polarity of the ions, add units to the y-axis, uniform the units of Dp (all nm or all m) and add a tick label near the cluster band.

35) Page 32 figure 2: the x-axis is and its label are confusing. Please make the x-axis so that have the same label and take the same space in this way the reader will be able to easily compare CS and cluster ions during the same month.

36) Figure 3, all panels: adding a visual indicator for the fog would help the reader to identify the fog period. Please consider adding it.

37) Figure 3, panel c): The secondary axis have a weird grey halo, please consider fixing it.

38) Figure 5, caption: "envolvement" pleases reword, this might not be English. Maybe "evolution"?

39) Figure 7, panel f): The Referee thinks that adding a label "D50 = 1.1e-7 m", or even better "D50 = 110 nm" instead of the number alone would make the figure easier to read.

40) Figure 9 and S3, all panels: please give only the significant digits for the fit.

---

## Author Comment (AC3) · 25 Sep 2017

**Responses to reviewer 3**

The changes in the manuscript are shown in italics here in the responses and they are marked in red in the revised manuscript.

Major comments:

**I) Page 9 line 16 to 21: The Referee is familiar with both methods to calculate the growth rates (mode fitting and appearance time) and still had hard time to follow the Authors' explanation. Please rephrase in a way that can be obvious also to the non- experts. Consider to add a figure in the supplementary material if the wording doesn't get any easier.**

Thanks to the reviewer for pointing this out. The description of the two growth rate determination methods is modified. No figures are added, because these methods have been presented in detail by Dal Maso et al. (2005) and Lehtipalo et al. (2014), respectively.

*In the mode-fitting method, at each time stamp of the measurement, the representative size of the aerosol population is determined by fitting a normal distribution to the measured concentration distribution along the logarithm of sizes with a base of 10. The mode of the fitted curve is transcribed back to linear scale and taken as the representative size of the particle population measured at this moment (a more detailed description of the method has been presented by Dal Maso et al., 2015 ). In contrast, in the appearance time method, for each size (the geometric mean size of a measurement size bin), one determines the time (the appearance time) at which the particle population is considered to reach this size, based on the measured concentration evolvement in time (Lehtipalo et al., 2014).*

**II) Page 12 line 8 to 10: "Owing ... the ionisation of air molecules can be neglected". The Referee thinks that this statement is highly speculative and needs to be modified. Are there measurements in the literature that support this claim? If so, please cite them. Furthermore, even if there is a snow pack, radon could make its way through the snow crystals and gamma radiation would need at least several tens of centimeter of solid ice to be shielded effectively. In addition, there can be areas, such as vertical walls that can be snow free and made of rock that is rich in uranium. If this is not the case, please state it in the paragraph with, if possible some references. Otherwise, rephrase the statement signaling its speculative nature.**

Our measurements were carried at Dome C on Antarctic plateau, which is the largest dissert on the earth covered by glaciers and snow. Solid ice and snow inhibit radon release at the South Pole (Maenhaut et al., 1979). The ice is even thicker at Dome C than at South Pole. The thickness of ice reported at Dome C by the European Project for Ice Coring in Antarctica (EPICA) is more than 3000 m (Augustin et al., 2004).

In continental air radon concentrations vary in the range of a few tens to hundreds of pCi m$^{-3}$ (Wilkniss and Larson, 1984). In the early 1970s at the South Pole, radon concentrations were in the order of 0.5 pCi m$^{-3}$ (Maenhaut et al., 1979), about the same or even less than over oceans (Wilkniss and Larson, 1984). The location of Dome C is much more inland than the South Pole back in the early 1970s. Therefore, characterised by the geographical location of Dome C, the contribution of radon or terrestrial gamma radiation to the ionisation in the air can be regarded non-existent. A slight modification is made to the sentence.

*Owing to the presence of the thick ice and snowpack (over 3000 m in depth; Augustin et al., 2004), the contribution of radon exhalation or terrestrial gamma emissions from ground at Dome C to the ionisation of air molecules can be neglected.*

**III) Page 13 line 1 to 6: As CS tended to be higher during NPF can author exclude that NPF was generated by condensable vapors from "polluted" air masses from the continent? The Referee thinks that a look at back trajectories is needed in order to understand weather the source of vapors is local or from long range transport. This information would be very valuable and increase the relevance of the paper.**
**& IV) Page 13 line 24 and 25: The Referee thinks that a look at back trajectories might help the Authors to be less speculative about the origin of the air mass. Therefore, the Referee feels strongly that a back trajectory analysis should be added and discussed.**

Response to III) & IV): The air entering Dome C originates almost entirely from subsidence from very high altitudes. Because of this, air mass back trajectories are not very helpful in tracking the sources of aerosols, or their precursors, at this site (air entering Dome C have plenty of ageing/mixing time during its transport from distant surface sources very difficult to capture with air mass trajectories). The situation would be totally different for coastal Antarctic sites, for which air mass back trajectories would be very helpful. We decided not to use back trajectories for estimating aerosol sources at Dome C, nor for estimating the reason for high values of CS during NPF events compared with event-free days during autumn.

**V) Chapter 3.2.2: A discussion on the uncertainties of the GR is completely lacking. The Referee thinks that must be added and discussed in relation to the variability of the GRs calculated with the 2 different methods.**

We are afraid that we have to point out that the uncertainties of GRs in relation to the procedures used in the two methods were discussed in the last paragraph in section 3.2.2.

*…could be ascribed to the higher uncertainties associated with the mode-fitting method. The mode-fitting method tracks the mode concentration corresponding to sizes based on curve fitting for each measurement cycle, and it could be that the sizes at which mode concentrations were identified apart significantly in two adjacent measurement cycles, i.e. over a short time interval. A large size difference over a small time interval, therefore, would lead to a huge instantaneous GR. In contrast, the*

*appearance time method is based on looking for the time stamp when the concentration reaches 75% of its maximum in the concentration vs. time space for each size channel of the instrument. Owing to the fact that aerosol and ion data have a higher resolution in the time dimension than in the size dimension, the appearance time method could pick up the time stamp more precisely for each size than the mode-fitting method could do the sizes for each measurement cycle. Consequently, the appearance time method presents GRs with smaller uncertainties (Fig. 5 & Table 1) and yields more representative instantaneous GRs.*

**VI) Page 16 line 23 to 26: The Referee thinks that an estimate of the uncertainty of J2+ is necessary in order to better assess the significance of the statistical parameters included in the text. In other words, is the standard deviation of the J2+ comparable, larger or smaller with respect to the uncertainties? Please add the missing information and discuss.**

The AIS typically underestimates the ion concentration by 15-30% in the size range of 2-3 nm (Wagner et al., 2016). We had no a separate CPC during the campaign measuring alongside the DMPS to estimate the uncertainty in the DMPS measurement. However, according to Wiedensohler et al. (2012), mobility particle size spectrometers of different design usually have an uncertainty range of around ± 10 % between 20 and 200 nm, and larger uncertainties are expected beyond this size range. In principle, the meteorological measurements at Concordia station follow WMO recommendations. According to WMO 2008 (WMO. Guide to meteorological instruments and methods of observation. Technical Report 8, World Meteorological Organisation, 2008.), the error for temperature measurement should be < 0.3 K in the range from -40 to +40 C and the uncertainty for pressure measurement should be about 0.1 hPa.

We made an estimation of uncertainties in J2+ by assuming an underestimation of 15-30% in our AIS measurement, an uncertainty of ± 10 % in our DMPS measurement in the whole size range, an error of ± 1 C in the temperature measurement and ± 1 hPa in the pressure measurement. We calculated the maximum and minimum estimates of J2+ based on these assumptions and evaluated the deviations of J2+ from the mean values. For more than 88% of the J2+ (26 cases in total), we obtained a deviation from the mean smaller than 0.020 cm-3s-1 (the standard deviation in all 26 J2+ values). More than 80% of the calculated deviation from the mean has a value smaller than 0.005 cm-3s-1.

*An estimation of uncertainties in $J_2^+$ was made by assuming an underestimation of 15-30% in the AIS measurement (Wagner et al., 2016), an uncertainty of ± 10 % in the DMPS measurement in the whole size range (Wiedensohler et al., 2012), an error of ± 1 C in the temperature measurement and ± 1 hPa in the pressure measurement. We calculated the maximum and minimum estimates of $J_2^+$ based on these assumptions and evaluated the deviations of $J_2^+$ from the mean values of the maximum and minimum estimates. We found that this deviation was smaller than 0.005 $cm^{-3}s^{-1}$ (<0.020 $cm^{-3}s^{-1}$) for more than 80% (88%) of the values of $J_2^+$.*

Minor comments:

Abstracts:
**1) Page1, line 24: "One ... days." Rephrase. It is unclear if the Authors want to refer to event days or event free days.**

We modified the sentence to deliver the message clearer that we refer to days with no sign of new particle formation, wind-induced ion formation, or ion production and loss associated with cloud/fog formation.

*For the subset of days when none of these processes seemed to operate, the concentrations of cluster ions (0.9-1.9 nm) exhibited a clear seasonality, with high concentrations in the warm months and low concentrations in the cold.*

**2) Page 2 Line 4 and 5: The Authors use 2 time the expression "work better" which is very vague. Please rephrase using a more precise terminology.**

The sentence is revised as follows

*The former method seemed to have advantages in characterising NPF events with a fast GR, whereas the latter method is more suitable when the GR was slow.*

**3) Page 2 Line 7 and 8: "The ion production ... seemed a unique feature at Dome C ..." Please be more specific. A unique feature in Dome C relative to the whole world? Relative to the other data collected in Antarctica?**

This refers to a feature that has not been reported for any other sites. The production of ions in the size range of 8-42 nm in relation to cloud formation therefore seems to be a unique feature at Dome C. The sentence is elaborated

*The ion production in relation to cloud/fog formation in the size range of 8-42 nm seemed to be a unique feature at Dome C, which has not been reported elsewhere.*

4**) Page 2 Line 10: "cleavage" replace with a word/rephrase using wording that is understandable to a wide audience.**

We replaced 'cleavage' with 'breakage'.

**3) Page 2 Line 11 and 12 "our observation ... ice crystals." This is a very interesting result! The Referee thinks that it should be explained more thoroughly in the appropriate paragraph (see comment XX)**

We believe that this result was presented and discussed in detail in section 3.3.2. Here in the abstract, we just provided a brief summary.

**Introduction 4) Page 2 line 17: Please add the size/mobility ranges next to "primary ions" and "aerosol particles" for a more complete information for the reader that might be new to atmospheric ions.**

Such information is added to the text.

*Air ions, also known as atmospheric ions, are electric charge carriers in the atmosphere, ranging from primary ions (most likely have a mobility diameter smaller than 0.8-1 nm) to charged aerosol particles (with a mobility diameter up to several hundred nm).*

**5) Page 3 line 2:"Such charged nanoparticles ... are typically observed during new particle formation (NPF) events." Confusing. Stable air ions are observed also without NPF. Please rephrase.**

The reviewer is right. This sentence is modified as
*Charged nanoparticles in the mobility size range of 1.7-7 nm are typically observed during new particle formation (NPF) events.*

**6) Page 3 line 9 to 20: "Carslaw et al. (2013) ... years (Fiebig et al., 2014)." Confusing paragraph. Please add a closing sentence that explains why this work is important and how it is different from the cited previous work in this way the list of previous papers on the topic will make more sense to the reader.**

Sorry for this confusion. This paragraph was meant to go together with the next paragraph. The information the reviewer looks for is actually contained in the following paragraph. These two paragraphs are combined into one in the revised manuscript.

**Methods 7) Page 5 line 4 and 5. Please add the flow rate after "high sample flow rate", which is too vague to be useful. The Referee is aware that there is a flow rate some 20 lines below, but the use of "High" is not helpful and repetita iuvant.**

The detailed flow rate information was given in the third paragraph of section 2.1.1. This first paragraph in section 2.1.1 was meant to serve an introductory purpose for section 2.1.1. However, since the review requested, we add the sample flow rate information as the reviewer suggested.

*The AIS employs two cylindrical multi-channel aspiration-type analysers and a high sample flowrate (60 l/min).*

**8) Page 7 line 21 to 23: The Referee thinks that a plot of the shifted spectrum added to the supplementary material could be very useful to the readers and AIS/NAIS users. I recommend adding it.**

Such a figure is added to the supplementary material.

**9) Page 9 line 4: "growth and coagulation". Remove coagulation.**

Coagulation can cause size increases of an aerosol population. Therefore, coagulation is not removed.

**10) Page 9 line 10: "Air ion and total aerosol particle data are three dimensional: " Remove. It is confusing and unnecessary.**

This sentence is removed as suggested.

**11) Page 9 line 12 and 13: the Authors refer to " mode fitting method" and "appearance time method" throughout as "former" and "latter". Please spell them out each time. Readers will have easier times at understanding to which method the Authors are referring to.**

These methods are spelt out as suggested.

**12) Page 9 line 26: "assist". Confusing term. Please rephrase.**

The word 'assist' is replaced by 'support'.

**13) Page 3 line 3: "Sulphuric acid is considered as a key chemical species". remove "as".**

The sentence is modified as

*Sulphuric acid is a key chemical species in forming aerosol particles in the ambient air.*

**14) Page 11 line 14:"During the campaign period, there were nearly 300 days with valid air ion measurements". Please add what is the corresponding percentage of valid days as well. The Referee is aware that the Authors refer to Table 1 where this information can be retrieve, but thinks that would be of help to the reader to make it explicit in the text.**

This information is given in the text.

*During the campaign period (330 days in total), there were 287 days with valid air ion measurements, i.e. valid air ion data were collected on nearly 87% of the measurement days.*

**15) Page 11 line 18: "definite". Confusing adjective. Please rephrase.**

The word 'definite' is replaced with 'certain'.

**16) Page 12 Line 2: "high" please add a value e.g., median or a range of warm months to help the reader.**

Typically, highest cluster ion concentrations were observed during warm months. Therefore, the sentence is modified as

*The cluster ion concentration was the highest during the warm months, with a maximum in February.*

**17) Page 12 line 4: "natural ionising radiation" is a confusing term as cosmic rays are also natural and ionizing. Please change wording.**

Natural ionizing radiation includes cosmic radiation. See the definition given by IAEA for example
https://www.iaea.org/newscenter/multimedia/photoessays/natural-and-artificial-ionizing-radiation

**18) Page 12 line 24: remove "Markedly" it is unnecessary. Please reword the sentence to make it more understandable/intuitive. In addition, do the Authors have information about the boundary layer (BL) in February, March, October and November? Can the Authors exclude that some of those high cluster ion concentration are due to limited mixing due to shallow BL? Please elaborate and include in the text.**

The word 'markedly' is removed as suggested and the sentence is modified.

*The daily-median cluster ion concentration at Dome C was observed to be higher on NPF event days compared with event-free days.*

Unfortunately, we have no information about the boundary layer height and we cannot exclude the shallow BL effect completely. However, the day length in February at Dome C is typically much longer compared with that in March and October. Therefore, a deeper mixed layer can be assumed in February than in March or October. However, the cluster ion concentration was found the highest in February. Therefore, even if the boundary layer height has an effect on the cluster ion concentration, this effect is likely to be very minor. A discussion about the boundary layer height effect is added in section 3.1 as follows

*The development of the planetary boundary layer may additionally influence the concentration of cluster ions by imposing either a dilution or concentration effect. The longer day length in February than in March or October may result in the development of a deeper mixed layer, which could dilute the cluster ions within the mixing volume. However, the highest cluster ion concentration was found in February. Also polar nights would cause the formation of only a very shallow and stable boundary layer in winter months. The mixing volume in winter therefore is expected much smaller than in other seasons, but no concentration effect on cluster ion concentration can be identified. Consequently, even if the seasonal change of boundary layer heights has an influence on the seasonality in cluster ion concentrations, this effect is likely to be minor.*

**19) Page 13 Line 20: "Bumps ... NPF events (Fig. 3c)". Please rephrase being clearer. Consider replacing the word "bumps" with e.g., "sudden increase".**

'bumps' is replaced with 'sudden increases' in the text as suggested by the reviewer.

**20) Page 13 line 24 and 25: "Such differences result … different origins".
Replace "differences" with "different".**

The sentence is modified as

*Such differences result likely from the availability of vapours that sustain the growth.*

**21) Page 13 line 26: replace "perceptible" with "measurable"**

The sentence is modified as

*We could see slight concentration increases in the cluster ion size range at the time
when NPF events were initiated, but…*

**22) Page 14 line 11: "Short after". Please be less vague, remove "short" and
add temporal information in the text.**

'Short after' is replace by 'About 4 hours after'

**23) Page 14 line 18 to 21: "slowly growing" … "slight growth". Please add
next to this general expression the value of the GR. It will make the text
more useful to the reader.**

The growth rate and size range information is added in the text for clearer
description. The wording 'slight' was a wrong interpretation. It is corrected in
the revised version.

*Over the consecutive five days on 12-16 February, a slowly-growing (GR ≈ 1.4 nm/h)
population of 40-200 nm particles could be observed in the background, with their
initial formation traceable back to 06:00 UTC on 12 February. Interestingly, apart
from the particles initiated at 10 nm and 40 nm, a third mode of particles with sizes
larger than 100 nm was recognisable on the morning of 12 February. This particle
mode grew approximately from 100 nm to 300 nm during 12-13 February, and then
gradually merged with the mode initiated at 40 nm at the end of 16 February.*

**24) Page 14 line 24: Replace "owing to" with "because of"**

We are thankful for he reviewer's suggestion. However, we decide to keep 'owing
to' as it is to avoid repeated usage of 'of'.

**25) Page 16 line 19: "… and yields more representative instantaneous GRs".
The word "representative" should be reserved to statistical analysis. If
such analysis was done to assess whether the calculated GRs were
statistically representative please add a sentence about the method used,
otherwise rephrase.**

We are afraid that we do not agree with the reviewer. In our opinion, the word 'representative' is not reserved to statistical analysis only. Here we made no statistical evaluation, but we decide to keep the word 'representative'.

**26) Page 16 line 20: The Referee thinks that a comparison between the GR for ions with Dp>10 nm and particles in the same size range should be discussed in this para- graph.**

We add discussion at the end of the last paragraph in section 3.2.2 as the reviewer suggested.

*At large sizes in the overlapping size range (10-42 nm) of AIS and DMPS, the instantaneous GRs derived from the AIS measurements tended to be larger than those from the DMPS measurements. This difference may result from the fact that the DMPS measures total particles, including both ions and neutral particles, whereas the AIS detects only charged particles. Also the AIS measurements at sizes larger than 20 nm are subject to the uncertainties brought by the detection of multiply charged particles as singly charged particles. At small sizes in the overlapping size range, the instantaneous GRs derived from the DMPS exhibited a decreasing trend with increasing sizes, which however was not shown by the instantaneous GRs derived from the AIS. This difference may again be attributed to the difference in the sampled particles targeted by the two instruments.*

**27) Page 17 line 17: "... 110 nm". Based on the data analyzed in this work and on literature is the activation in this size range typical? The Referee recommends, if possible, adding some discussion about this in this paragraph.**

As suggested by the referee, we modified this paragraph into the following form:

*This observation is well in line with the activation thresholds from <50 nm up to about 200-300 nm for the "dry" particle diameter observed in real atmospheric clouds (see Henning et al., 2002, and references therein; Anttila et al., 2009; Kyrö et al., 2013; Portin et al., 2014; Leaitch et al., 2016). Clusters ions were efficiently lost onto the cloud droplets at Dome C.*

**28) Page 18 line 11: "we observed wind-induced ion formation especially during the winter months". Please add some numbers to make this statement less vague. How many times in winter with respect to other months?**

The sentence is modified by taking in to account the reviewer's comment as

*We observed wind-induced ion formation especially during the dark months (15 cases during May-August).*

**29) Page 18 line 24: " Ionising radiation produces primary ions via ionisation". Please either remove "via ionization" or add what is ionized e.g., "via ionization of vapor molecules"**

'via ionisation' is removed according to the reviewer's suggestion.

**30) Page 19 line 8: "However ... contribute to the ion burst captured by the AIS". The Referee thinks that this statement is highly speculative and unsubstantiated. The Authors should discuss more in length adding references.**

The discussion on this point was actually given in the following up text in this paragraph. In this paragraph, we proposed a mechanism that we think might be the cause to the ion formation during strong wind episodes, which needs further experimental validation as mentioned in the text. The text in the beginning of the paragraph is modified to better show the speculative nature of the whole paragraph.

*Turbulent conditions might enhance the collection of electric charges by the shattered snowflakes and ice particles via a charge transfer from initial charge carriers, contributing to the formation of an ion burst. In addition, the shattered particles might gain electric charges through friction charging. However, we think that these two pathways of ion formation are not likely to contribute to the ion burst captured by the AIS. In principle,...*

**31) Page 19 line 22: "similar" the wording makes the sentence too generic and vague, please discuss further, how those feature are similar.**

The sentence is modified as

*By putting together all the 36 wind-induced ion formation events, a linear correlation was identified between the logarithm of the ion concentration and wind speed (Fig. 9), like also found at Aboa.*

**32) Page 20 line 3: "unexpectedly" unnecessary adjective, please remove.**

The word 'unexpectedly' is removed as suggested by the reviewer.

**33) All figures of DMPS and AIS size distributions: add units to dN/dlogDp make the units of Dp consistent (all nm or all m) add thick labels so that are consistent and at least 2 in number e.g., 10 and 100 nm**

units are added to dN/dlogdp and dp are shown in nm and tick labels are added as reviewer suggested to all DMPS and AIS contour plots.

**34) Page 31 caption of figure 1: specify the polarity of the ions, add units to the y-axis, uniform the units of Dp (all nm or all m) and add a tick label near the cluster band.**

The polarity is specified and the unit is added to the y-axis label. Dp are shown in nm and a tick label is added in the contour plot to point on the cluster band. The modified Figure 1 and its captions are

[Figure]

*Figure 1. The median size distribution of* positive *ions measured by the AIS on an event-free day (16 January, 2011). The measured number size distribution of this day is shown in the contour plot.*

**35) Page 32 figure 2: the x-axis is and its label are confusing. Please make the x-axis so that have the same label and take the same space in this way the reader will be able to easily compare CS and cluster ions during the same month.**

Figure 2 is fixed according to the reviewer's suggestion.

[Figure]

Figure 2. Seasonality in the median a) cluster ion (0.9-1.9 nm) concentration and b) condensation sink (CS). Tops and bottoms of the boxes are the 75th and 25th percentiles of the median daily values in 10 min time resolution, with bars in the middle showing the 50th percentiles. Whiskers represent spans of the interquartile ranges multiplied by 1.5. Cluster ion concentrations or CS on new particle formation (NPF) days shown in red and on event-free days in black. Event-free conditions were restricted to days, on which no NPF, cloud activation, wind-induced events or contamination as well as other anomalies altering the ion concentration in the cluster band. The numbers of days classified as either event-free or NPF are displayed on the top of the panel a) in grey colour. No CS was obtained in August due to the lack of measured temperature and pressure data from the station database.

**36) Figure 3, all panels: adding a visual indicator for the fog would help the reader to identify the fog period. Please consider adding it.**

No fog was observed during the time shown in Figure 3.

**37) Figure 3, panel c): The secondary axis have a weird grey halo, please consider fixing it.**

The grey halo is removed.

**38) Figure 5, caption: "envolvement" pleases reword, this might not be English. Maybe "evolution"?**

We used 'evolvement', which is an English word. However, it is changed to 'evolution' as the reviewer suggested.

**39) Figure 7, panel f): The Referee thinks that adding a label "D50 = 1.1e-7 m", or even better "D50 = 110 nm" instead of the number alone would make the figure easier to read.**

Figure 7f is modified according to the reviewer's suggestion.

[Figure]

**40) Figure 9 and S3, all panels: please give only the significant digits for the fit.**

The fitting parameters are rounded to 2 significant digits and are presented in Table S1 in the revised manuscript.

*Table S1. Coefficients for the fittings shown in Figs. 9 and S4. $R^2$ is the coefficient of determination measuring the goodness of fit, which denotes the fraction of the total variation in the data can be explained by the fit. For Dome C data shown in Fig. 9, fits 1 and 2 are obtained based on all data below or above the wind speed threshold (7 m/s), respectively. The grey data points in Fig. 9 are used in determining the fitting coefficients for fits 3 and 4. For Aboa data shown in Fig. S4, a wind speed threshold of 17 m/s is used.*

| | | | | | | |
|---|---|---|---|---|---|---|
| | *Cluster (0.9-1.9 nm) ion concentrations vs. wind speeds* | | | | | |
| | *Fits* | *a* | *b* | *95% conference interval for a* | *95% conference interval for b* | *$R^2$* |
| | *1* | *0.69* | *26.34* | *[0.65 0.73]* | *[21.62 32.10]* | *0.24* |
| | *2* | *0.51* | *68.64* | *[0.41 0.60]* | *[29.88 157.67]* | *0.12* |
| *DOME C (Fig. 9)* | *3* | *0.73* | *21.83* | *[0.69 0.77]* | *[18.02 26.44]* | *0.28* |
| | *4* | *0.44* | *327* | *[0.40 0.47]* | *[244.95 436.53]* | *0.52* |
| | *1.9-10 nm ion concentrations vs. wind speeds* | | | | | |
| | *Fits* | *a* | *b* | *95% conference interval for a* | *95% conference interval for b* | *$R^2$* |
| | *1* | *1.14* | *0.07* | *[1.08 1.2]* | *[0.05 0.09]* | *0.29* |
| | *2* | *0.61* | *2.1* | *[0.71 0.88]* | *[0.88 5.01]* | *0.15* |
| | *3* | *1.19* | *0.05* | *[1.25 0.04]* | *[0.04 0.07]* | *0.31* |
| | *4* | *0.54* | *9.87* | *[0.58 7.18]* | *[7.18 13.58]* | *0.58* |
| | *0.9-2.2 nm ion concentrations vs. wind speeds* | | | | | |
| | *Fits* | *a* | *b* | *95% conference interval for a* | *95% conference interval for b* | *$R^2$* |
| *ABOA (Fig. S4)* | *6* | *0.17* | *14.98* | *[0.15 0.19]* | *[9.02 24 88]* | *0.66* |
| | *2.2-9.5 nm ion concentrations vs. wind speeds* | | | | | |
| | *Fits* | *a* | *b* | *95% conference interval for a* | *95% conference interval for b* | *$R^2$* |
| | *5* | *0.24* | *1.45* | *[0.22 0.26]* | *[1.26 1.68]* | *0.35* |
| | *6* | *0.06* | *63.24* | *[0.04 0.09]* | *[34.72 115.18]* | *0.17* |

References:

Augustin, L., Barbante, C., Barnes, P. R. F., Barnola, J. M., Bigler, M., Castellano, E., Cattani, O., Chappellaz, J., Dahl-Jensen, D., Delmonte, B., Dreyfus, G., Durand, G., Falourd, S., Fischer, H., Flückiger, J., Hansson, M. E., Huybrechts, P., Jugie, G., Johnsen, S. J., Jouzel, J., Kaufmann, P., Kipfstuhl, J., Lambert, F., Lipenkov, V. Y., Littot, G. C., Longinelli, A., Lorrain, R., Maggi, V., Masson-Delmotte, V., Miller, H., Mulvaney, R., Oerlemans, J., Oerter, H., Orombelli, G., Parrenin, F., Peel, D. A., Petit, J.-R., Raynaud, D., Ritz, C., Ruth, U., Schwander, J., Siegenthaler, U., Souchez, R., Stauffer, B., Steffensen, J. P., Stenni, B., Stocker, T. F., Tabacco, I. E., Udisti, R., Wal, R. S. W. v. d., Broeke, M. v. d., Weiss, J., Wilhelms, F., Winther, J.-G., Wolff, E. W., and Zucchelli, M.: Eight glacial cycles from an Antarctic ice core, Nature, 429, 623-628, 2004.

Dal Maso, M., Kulmala, M., Riipinen, I., Wagner, R., Hussein, T., Aalto, P. P., and Lehtinen, K. E. J.: Formation and growth of fresh atmospheric aerosols: eight years of aerosol size distribution data from SMEAR II, Hyytiälä, Finland, Boreal Env. Res., 10, 323 - 336, 2005.

Lehtipalo, K., Leppä, J., Kontkanen, J., Kangasluoma, J., Franchin, A., Wimmer, D., Schobesberger, S., Junninen, H., Petäjä, T., Sipilä, M., Mikkilä, J., Vanhanen, J., Worsnop, D. R., and Kulmala, M.: Methods for determining particle size distribution and growth rates between 1 and 3 nm using the Particle Size Magnifier, Boreal Environ. Res., 19 (suppl. B), 215-236, 2014.

Maenhaut, W., Zoller, W. H., and Coles, D. G.: Radionuclides in the south pole atmosphere, Journal of Geophysical Research, 84, 3131, 10.1029/JC084iC06p03131, 1979.

Wagner, R., Manninen, H. E., Franchin, A., Lehtipalo, K., Mirme, S., Steiner, G., Petäjä, T., and Kulmala, M.: On the accuracy of ion measurements using a Neutral cluster and Air Ion Spectrometer, Boreal Env. Res., 21, 230-241, 2016.

Wiedensohler, A., Birmili, W., Nowak, A., Sonntag, A., Weinhold, K., Merkel, M., Wehner, B., Tuch, T., Pfeifer, S., Fiebig, M., Fjäraa, A. M., Asmi, E., Sellegri, K., Depuy, R., Venzac, H., Villani, P., Laj, P., Aalto, P., Ogren, J. A., Swietlicki, E., Williams, P., Roldin, P., Quincey, P., Hüglin, C., Fierz-Schmidhauser, R., Gysel, M., Weingartner, E., Riccobono, F., Santos, S., Grüning, C., Faloon, K., Beddows, D., Harrison, R., Monahan, C., Jennings, S. G., O'Dowd, C. D., Marinoni, A., Horn, H. G., Keck, L., Jiang, J., Scheckman, J., McMurry, P. H., Deng, Z., Zhao, C. S., Moerman, M., Henzing, B., Leeuw, G. d., Löschau, G., and Bastian, S.: Mobility particle size spectrometers: harmonization of technical standards and data structure to facilitate high quality long-term observations of atmospheric particle number size distributions, Atmos. Meas. Tech., 5, 657-685, 10.5194/amt-5-657-2012, 2012.

Wilkniss, P. E., and Larson, R. E.: Atmospheric radon measurements in the Arctic; Fronts, seasonal observations, and transport of continental air to polar regions, J. Atmos. Sci., 41, 2347-2358, 1984.

---

## Author Response (AR1)

We are thankful to the careful reviews provided by the referees. Responses to the referees' comments are presented below. The changes made in the manuscript are shown in italics here in the responses and they are marked in red in the revised manuscript.

**Responses to reviewer 1**

**1.      Abstract (page 2, lines 11-13): Odd sentence, consider revision.**

The sentence is revised as follows

*For the wind-induced ion formation, our observations suggest that the ions originated more likely from atmospheric nucleation of vapours released from the snow than from mechanical charging of shattered snow flakes and ice crystals.*

**2.      Introduction (page 4). For non-experts, I especially miss a short statement about cosmic ray intensity at Dome C compared to mid-latitudes. In addition: Is the total intensity of ionising radiation comparable to continental mid-latitudes?**

We are thankful that the reviewer pointed this out. A short statement as the reviewer suggested is added in the introduction on page 4 on lines 3-4 as follows:

*Also stronger cosmic ray ionisation can be expected at polar regions than mid-latitudes (Kazil et al., 2006; Bazilevskaya et al., 2008).*

In principle, at Dome C, the only source of ionising radiation comes from cosmic radiation whereas at continental mid-latitudes in addition to cosmic radiation, there are also the contributions to ionising radiation from the decay of radon and other radioactive nuclides. However, the high altitude of Dome C can make us speculate that the intensity of cosmic radiation is higher at Dome C than at continental mid-latitudes. But since we had no measurements of ionising radiation during this campaign, it is not possible to tell whether the total intensity of ionising radiation is comparable to continental mid-latitudes or not.

**3.      Methods (page 4, lines 16-19): I wonder if the experiments were installed in a separate hut somewhat upwind to the main station as described in Järvinen et al. (2013) or Becagli et al. (2012). In this regard, the authors should briefly address the potential problem of local contamination.**

The sampling site is the same as was used by Udisti et al. (2012) and Becagli et al. (2012) for taking filter samples and by Järvinen et al. (2013) for measuring particle number size distributions with a DMPS. The site is located about 1 km southwest of the station main buildings, upwind in the direction of the prevailing wind. The northeastern direction was declared as the contaminated sector (10–90°) due to diesel generator and motor vehicle emissions at the station. Contaminated data was very clearly visible as high and noisy concentrations in particle size ranges > 10 nm simultaneously with the AIS and the DMPS  and also as high absorption coefficients with the Particle Soot Absorption Photometer (PSAP) (Grythe, 2017). Consequently, the data were omitted from further analysis when the measured winds were from the contaminated sector. The description of the site is elaborated.

*Measurements were installed at the same sampling site used by Järvinen et al. (2013) and Becagli et al. (2012), which located upwind in the direction of the prevailing wind at a distance of about 1 km southwest of the main station buildings. The northeastern direction is considered as the contaminated sector (10–90°), due to local emissions from diesel generators and motor vehicles.*

**4.      Chapter 3.2.1 (page 13): Is it possible to assess the impact of neutral clusters on NPF? Definitely, this may be an additional important issue for dedicated future investigations at this site (see Conclusions, page 21, last section).**

Thanks to the reviewer for bring up this issue. Indeed, it would be very interesting to understand the relative importance of neutral and charged clusters in NPF. This is an on-going topic in atmospheric NPF studies and it should be one aspect to look further into at Dome C in the future. However, during the 2010-2011 campaign, we had no measurement on neutrals, neither are we able to derive the neutral fractions of clusters based on the set of instrumentations (a DMPS and an AIS) we had. However, we elaborate the last paragraph in the Conclusions with a brief discussion on the need for assessing the role of neutrals in NPF.

*In the future in addition to air ions, also the properties of neutral clusters and particles need to be probed in order to understand the relative importance of ions and neutrals in atmospheric NPF at Dome C, and to characterise the comparability of the roles of ions and neutrals in atmospheric NPF observed at Dome C and at other sites around the globe.*

**5.      Chapter 3.2.2 and 3.2.3: Evaluation of particle growth rates and ion formation rates as described presupposes that NPF occurred in homogeneous air masses, thus neglecting the potential role transport and mixing processes. I think it is worthwhile to allude to (and discuss) the impact of these processes on the variance of particle growth- and ion formation rates, especially in regard with the derived extraordinarily high instantaneous growth rates up to some 100 nm/h.**

We indeed assumed that the air was homogenously mixed. According to early-published work on NPF classification based on ion spectrometers (Hirsikko et al., 2007; Manninen et al., 2010), these 'banana' shape events as shown in our Figure 5 are typically regional phenomena over a large area, within which the air has a homogeneous characteristics, implied by the smooth growth over several hours (Hirsikko et al., 2007). Therefore, the influence from transport and mixing processes are assumed negligible. A sentence is added to the first paragraph in section 3.2.1 concerning this issue as follows

*The smooth growth that lasts for several hours can imply a homogeneous condition in the sampled air (Hirsikko et al., 2007; Manninen et al., 2010).*

The very high instantaneous GRs likely mainly originate from uncertainties in the GR determination methods in treating the number size distribution data as discussed in the last paragraph in section 3.2.2, due to the fact that the number size distribution data has a much better resolution in time than in sizes.

**6.      Chapter 3.3.2 and Figure 9: In the Introduction, the particular conditions at Dome C, i.e. pronounced ionisation rates but limited source of vapours for clustering, were stressed (page 4, lines 3-7). Looking at Fig. 9a, I realized that ion concentrations in the size range 0.9 – 1.9 nm were roughly about a few hundred at low wind speeds (< 5m/s), which is in turn comparable to ion concentrations observed in a boreal forest (Chen et al., 2016; Figure 9 therein, sum over size range 0.8 – 1.7 nm), a site where sources of condensable vapours should not be limited. Does this analogy indicate that higher ionisation counterbalanced the lack of condensable vapours at Dome C? I think it may be worthwhile to speculate about this point.**

The reviewer's suggestion is appreciated. At SMEAR II station, both the decay of radioactive nuclides and cosmic radiation contribute to ionisation. At Dome C however, due to the presence of the thick glaciers, the terrestrial radioactivity hardly can contribute to ionisation in the atmosphere. That is, there are less sources of ionising radiation at Dome C compared with SMEAR II station. Although the cosmic radiation intensity can be expected to be higher at Dome C than at SMEAR II, since we had no ionising radiation measurements for the 2010-2011 campaign, it is not possible to tell whether the ionisation is higher or not at Dome C compared with that at SMEAR II station (the measurement site in Chen et al. 2016). Therefore, it is not feasible to speculate as the reviewer suggested.

**7.      Table 1: This table is redundant and can readily be removed, because it provides no further information as already presented in the main text (Chapter 3.1, page 11). If at all, a plot showing the occurrence of the different features during the observation period on a time scale could be much more enlightening.**

Table 1 is removed from the manuscript as suggested by the reviewer, but no plot is added.

References:

Bazilevskaya, G. A., Usoskin, I. G., Flückiger, E. O., Harrison, R. G., Desorgher, L., Bütikofer, R., Krainev, M. B., Makhmutov, V. S., Stozhkov, Y. I., Svirzhevskaya, A. K., Svirzhevsky, N. S., and Kovaltsov, G. A.: Cosmic Ray Induced Ion Production in the Atmosphere, Space Sci Rev 137, 149–173, 10.1007/s11214-008-9339-y, 2008.

Becagli, S., Scarchilli, C., Traversi, R., Dayan, U., Severi, M., Frosini, D., Vitale, V., Mazzola, M., Lupi, A., Nava, S., and Udisti, R.: Study of present-day sources and transport processes affecting oxidised sulphur compounds in atmospheric aerosols at Dome C (Antarctica) from year-round sampling campaigns, Atmospheric Environment, 52, 98-108, 10.1016/j.atmosenv.2011.07.053, 2012.

Grythe, H.: Quantification of sources and removal mechanisms of atmospheric aerosol particles. , PhD, Department of Environmental Science and Analytical Chemistry, Stockholm University, 2017.

Hirsikko, A., Bergman, T., Laakso, L., Maso, M. D., Riipinen, I., Hõrrak, U., and Kulmala, M.: Identification and classification of the formation of intermediate ions measured in boreal forest, Atmos. Chem. Phys., 7, 201–210, 2007.

Järvinen, E., Virkkula, A., Nieminen, T., Aalto, P. P., Asmi, E., Lanconelli, C., Busetto, M., Lupi, A., Schioppo, R., Vitale, V., Mazzola, M., Petäjä, T., Kerminen, V. M., and Kulmala, M.: Seasonal cycle and modal structure of particle number size distribution at Dome C, Antarctica, Atmospheric Chemistry and Physics, 13, 7473-7487, 10.5194/acp-13-7473-2013, 2013.

Kazil, J., Lovejoy, E. R., Barth, M. C., and O'Brien, K.: Aerosol nucleation over oceans and the role of galactic cosmic rays, Atmos. Chem. Phys., 6, 4905-4924, 2006.

Manninen, H. E., Nieminen, T., Asmi, E., Gagné, S., Häkkinen, S., Lehtipalo, K., Aalto, P. P., Vana, M., Mirme, A., Mirme, S., Hõrrak, U., Plass-Dülmer, C., Stange, G., Kiss, G., Hoffer, A., Törő, N., Moerman, M., Henzing, B., Leeuw, G. d., Brinkenberg, M., Kouvarakis, G. N., Bougiatioti, A., Mihalopoulos, N., O'Dowd, C., Ceburnis, D., Arneth, A., Svenningsson, B., Swietlicki, E., Tarozzi, L., Decesari, S., Facchini, M. C., Birmili, W., Sonntag, A., Wiedensohler, A., Boulon, J., Sellegri, K., Laj, P., Gysel, M., Bukowiecki, N., Weingartner, E., Wehrle, G., Laaksonen, A., Hamed, A., Joutsensaari, J., Petäjä, T., Kerminen, V. M., and

Kulmala, M.: EUCAARI ion spectrometer measurements at 12 European sites – analysis of new particle formation events, Atmos. Chem. Phys., 10, 7907-7927, 10.5194/acp-10-7907-2010, 2010.

Udisti, R., Dayan, U., Becagli, S., Busetto, M., Frosini, D., Legrand, M., Lucarelli, F., Preunkert, S., Severi, M., Traversi, R., and Vitale, V.: Sea spray aerosol in central Antarctica. Present atmospheric behaviour and implications for paleoclimatic reconstructions, Atmospheric Environment, 52, 109-120, 10.1016/j.atmosenv.2011.10.018, 2012.

**Responses to reviewer 2**

**1. Page 3., line 27: remove "totally"**

The word 'totally' is removed as suggested by the reviewer.

**2. Section Introduction: MS deals with polar region, however a brief summary or overview on similarities/differencies with Arctic region would be advantegous.**

In this manuscript, Antarctica is studied because of its remoteness from anthropogenic sources, not because of its polar character. In this respect, Antarctic differs substantially from Arctic areas, the latter of which are exposed strongly to anthropogenic sources outside much of the summer season. We feel that discussing the similarities/differences between Antarctica and Arctic might confuse the readers, rather than be helpful in putting the results of this paper into a larger context.

**3. Section Introduction: more specific information on cosmic radiation needed**

A short statement is added in the introduction on page 4 lines 3-4 as suggested by reviewer 1 regarding the ionisation by cosmic radiation as follows:

*Also stronger cosmic ray ionisation can be expected at polar regions than mid-latitudes (Kazil et al., 2006; Bazilevskaya et al., 2008).*

**4.        Section 2.1: the description of AIS speaks for itself, but there is no information about diffusion losses what is a relevant question for nanoparticles. How were the sampling lines set up? How long were they? How were the diffusional losses taken into account?**

The AIS had a separate copper inlet tube through the wall of the measurement container, similar to the one used by Virkkula et al. (2007) at Aboa. It was 30 cm long and its inner diameter was 16 mm. The volume of the inlet was thus ~0.06 L and with the inlet flow of 60 LPM the residence time within the inlet tube was 60 ms. The diffusional losses of ions inside the AIS was taken into account in the data inversion software designed for the AIS (Mirme et al., 2007). However, we did not take into account the possible diffusional loss of the inlet, because a reliable diffusional loss correction requires the knowledge of the temperature profile at the inlet. The AIS sat in a cabin of room temperature, but the inlet was extended outside of the cabin. We had no temperature measurement at the sampling inlet. Including a diffusional loss correction based on assumptions for inlet temperatures would actually introduce uncertainties to the measured number size distributions of air ions. Therefore, we decided to report the data without inlet diffusional loss correction. This issue is explained in the revised manuscript in the second last paragraph of section 2.1.1. as

*The deployed AIS had a separate 30 cm long inlet that extended outside the measurement cabin. The inlet tube had an inner diameter of 16 mm. However, since we had no measurement of the temperature profile of the inlet, a correction for the inlet diffusional loss is not feasible. Therefore, we report the number size distribution data of air ions without the inlet diffusional loss correction.*

**5.        Page 9., line 5: The units have to standardized in the paper, and the form of "nm h-1" should be preferred instead of "nm/h".**

'nm h-1' is now used in the manuscript as the reviewer suggested.

**6.    Page 10., equation 7: remove the integration limits 0 and infinite (see Dal Maso et al., 2005)**

Limits are removed as the reviewer suggested.

**7.    Page 10., line 22: Was the dry condensation sink calculation used? What about RH dependency?**

Yes, we used the dry condensation sink. Under moisture conditions, the application of the hygroscopy correction can increase the condensation sink, which is important to be taken into account for the determination of the amount of condensable vapour source. However in this study, the condensation sink was used as a proxy for aerosol loadings in the atmosphere, where the RH effect is not crucial. Also dome C is rather cold, which makes it a relatively dry place.

**8.    Page 11., line 15: Summary on classification of the measurement days would help to better understand the distinction of the days, and thus the description of Table 1. has to be shortened**

We decided to remove Table 1 from the manuscript completely according to the comment of reviewer 1. The information contained in Table 1 is given in the text.

**9. Page 15., 1. paragraph: Repetition from earlier.**

The methods for the growth rate determination are briefly repeated in this paragraph to help the readers to recall the difference between them. We decided to leave this paragraph as it is, because we think that this paragraph can assist readers in understanding the features in GRs presented in the following text in section 3.2.2.

**10. Page 16. line 4: The interpretation of intervals has to standardized in the paper, e.g. instead of "0.5 – 25", "0.5–25" should be used everywhere.**

The interval expression is fixed in the manuscript as suggested by the reviewer.

**11. Page 11., line 3: Is there any possible reason why the ion formation rates are comparable to those environments? Any comments regarding to altitudes?**

Dome C typically has much smaller condensation sinks (CS) than the sites reported by Manninen et al. (2010) and Nieminen et al. (2011). CS at Dome was in the order of $10^{-4}$ $cm^{-3}s^{-1}$ (Järvinen et al., 2013), whereas CS reported for the other site were in the order of $10^{-3}$ $cm^{-3}s^{-1}$ (Manninen et al., 2010; Nieminen et al., 2011). A low CS means a low uptake of vapour and electric charges on the aerosol particles. Although vapour sources are limited at Dome C, the availability of vapours in the air for nucleation and growth may be comparable to that at the other sites due to the low CS. The availability of vapours is the most essential factor regarding ion formation in the size range of 2-3 nm.

The high altitude of Dome C would mean a higher exposure to cosmic radiation and therefore a high contribution to ionisation in the atmosphere by comic radiation. However, since there was no ionising radiation measurement, it is not possible to tell how comparable the overall ionising radiation level at Dome C is to that at other sites, which makes it impossible to deduce the role of enhanced cosmic ray ionisation at the high altitude in relation to ion formation rates.

**12. Page 39, Fig. 9: Ionising radiation as third variable (colored circles) could be added to the plot. Also, the non-linear relation is evident at least in case of Fig. 9b.**

We appreciate that reviewer's suggestion. However, we had no ionising radiation measurement during the 2010-2011 campaign. Therefore, it is not possible to have ionising radiation plotted as a third variable in Fig. 9.

The reviewer could be right that there is a two-step linear relationship between the logarithm of the 1.9-10 nm ion concentration and wind speed, which seems to also exist between the cluster ion concentration and wind speed. The description in the manuscript regarding this feature is revised and also the fittings in Figs. 9 and S4 are updated accordingly. In the revised manuscript, all fitting parameters are presented in Table S1.

*By putting together all the 36 wind-induced ion formation events, the logarithm of the ion concentration exhibited linear relations to the wind speed (Fig. 9), like also observed at Aboa (Virkkula et al., 2007). For both cluster ions and ions in the size range of 1.9-10 nm, there seemed to be a two-step linear relation with a breakpoint at around 7 m s$^{-1}$ (Fig. 9). Winds below this threshold value were less efficient in producing ions than winds with speeds > 7 m s$^{-1}$. This feature could be also recognised in the Aboa data, but with the threshold in wind speeds lying at around 17 m s$^{-1}$ (Fig. S4). The effect of wind on ions seemed to be stronger at Dome C than at Aboa (Table S1 and Fig. S4).*

[Figure]

Figure 9. Ion concentrations as a function of wind speeds. a) ion concentration in the cluster size range (0.9-1.9 nm) and b) ion concentration in the size range of 1.9 and 10 nm. The solid lines are linear fits to the data. Fits 1 and 3 are to data with a wind speed below the threshold wind speed (7 m/s) and Fits 2 and 4 are to data above the threshold wind speed. Fits 1 and 2 are obtained based on all data below or above the wind speed threshold, respectively. The data points in grey colour, however, are not taken into account in determining the fitting coefficients for fits 3 and 4. These grey data points correspond to cluster ion concentration values below the purple dashed line ($y = 0.0074 \cdot e^{1.4855x}$). The coefficients of these fits as well as the 95% confidence bounds of the coefficients and coefficient of determination measuring the goodness of fit are shown in Table S1.

[Figure]

Figure S4. Ion concentrations as a function of wind speeds. a) Ion concentration in the cluster size range: 0.9-1.9 nm for Dome C (black circles) and 0.9-2.2 nm for Aboa (grey circles, from Virkkula et al. (2007)). b) Ion concentration in the size range of 1.9-10 nm for Dome C (black circles) and in the intermediate size range of 2.2-9.5 nm for Aboa (grey circles, from Virkkula et al. (2007)). The Aboa ion data were reported in mass diameters. The size ranges referred here are reconverted from the measured electrical mobility channels in mobility diameters. The solid lines are linear fits to the logarithm of the ion concentration data. The fitting parameters are given in Table S1. A wind speed threshold of 17 m/s is used for characterising the 2-step linear feature.

Table S1. Coefficients for the fittings shown in Figs. 9 and S4. $R^2$ is the coefficient of determination measuring the goodness of fit, which denotes the fraction of the total variation in the data can be explained by the fit. For Dome C data shown in Fig. 9, fits 1 and 2 are obtained based on all data below or above the wind speed threshold (7 m/s), respectively. The grey data points in Fig. 9 are used in determining the fitting coefficients for fits 3 and 4. For Aboa data shown in Fig. S4, a wind speed threshold of 17 m/s is used.

| | | | | Cluster (0.9-1.9 nm) ion concentrations vs. wind speeds | | |
|---|---|---|---|---|---|---|
| | Fits | $a$ | $b$ | 95% conference interval for $a$ | 95% conference interval for $b$ | $R^2$ |
| | 1 | 0.69 | 26.34 | [0.65 0.73] | [21.62 32.10] | 0.24 |
| | 2 | 0.51 | 68.64 | [0.41 0.60] | [29.88 157.67] | 0.12 |
| | 3 | 0.73 | 21.83 | [0.69 0.77] | [18.02 26.44] | 0.28 |
| | 4 | 0.44 | 327 | [0.40 0.47] | [244.95 436.53] | 0.52 |
| | | | | 1.9-10 nm ion concentrations vs. wind speeds | | |
| | Fits | $a$ | $b$ | 95% conference interval for $a$ | 95% conference interval for $b$ | $R^2$ |
| | 1 | 1.14 | 0.07 | [1.08 1.2] | [0.05 0.09] | 0.29 |
| | 2 | 0.61 | 2.1 | [0.71 0.88] | [0.88 5.01] | 0.15 |
| | 3 | 1.19 | 0.05 | [1.25 0.04] | [0.04 0.07] | 0.31 |
| | 4 | 0.54 | 9.87 | [0.58 7.18] | [7.18 13.58] | 0.58 |
| | | | | 0.9-2.2 nm ion concentrations vs. wind speeds | | |
| | Fits | $a$ | $b$ | 95% conference interval for $a$ | 95% conference interval for $b$ | $R^2$ |

DOME C (Fig. 9)

ABO

| | | | | | |
|---|---|---|---|---|---|
| 6 | 0.17 | 14.98 | [0.15 0.19] | [9.02 24 88] | 0.66 |
| 2.2-9.5 nm ion concentrations vs. wind speeds | | | | | |
| Fits | $a$ | $b$ | 95% conference interval for $a$ | 95% conference interval for $b$ | $R^2$ |
| 5 | 0.24 | 1.45 | [0.22 0.26] | [1.26 1.68] | 0.35 |
| 6 | 0.06 | 63.24 | [0.04 0.09] | [34.72 115.18] | 0.17 |

**III) Page 13 line 1 to 6: As CS tended to be higher during NPF can author exclude that NPF was generated by condensable vapors from "polluted" air masses from the continent? The Referee thinks that a look at back trajectories is needed in order to understand weather the source of vapors is local or from long range transport. This information would be very valuable and increase the relevance of the paper.**
**& IV) Page 13 line 24 and 25: The Referee thinks that a look at back trajectories might help the Authors to be less speculative about the origin of the air mass. Therefore, the Referee feels strongly that a back trajectory analysis should be added and discussed.**

Response to III) & IV): The air entering Dome C originates almost entirely from subsidence from very high altitudes. Because of this, air mass back trajectories are not very helpful in tracking the sources of aerosols, or their precursors, at this site (air entering Dome C have plenty of ageing/mixing time during its transport from distant surface sources very difficult to capture with air mass trajectories). The situation would be totally different for coastal Antarctic sites, for which air mass back trajectories would be very helpful. We decided not to use back trajectories for estimating aerosol sources at Dome C, nor for estimating the reason for high values of CS during NPF events compared with event-free days during autumn.

**V) Chapter 3.2.2: A discussion on the uncertainties of the GR is completely lacking. The Referee thinks that must be added and discussed in relation to the variability of the GRs calculated with the 2 different methods.**

We are afraid that we have to point out that the uncertainties of GRs in relation to the procedures used in the two methods were discussed in the last paragraph in section 3.2.2.

*…could be ascribed to the higher uncertainties associated with the mode-fitting method. The mode-fitting method tracks the mode concentration corresponding to sizes based on curve fitting for each measurement cycle, and it could be that the sizes at which mode concentrations were identified apart significantly in two adjacent measurement cycles, i.e. over a short time interval. A large size difference over a small time interval, therefore, would lead to a huge instantaneous GR. In contrast, the appearance time method is based on looking for the time stamp when the concentration reaches 75% of its maximum in the concentration vs. time space for each size channel of the instrument. Owing to the fact that aerosol and ion data have a higher resolution in the time dimension than in the size dimension, the appearance time method could pick up the time stamp more precisely for each size than the mode-fitting method could do the sizes for each measurement cycle. Consequently, the appearance time method presents GRs with smaller uncertainties (Fig. 5 & Table 1) and yields more representative instantaneous GRs.*

**VI) Page 16 line 23 to 26: The Referee thinks that an estimate of the uncertainty of J2+ is necessary in order to better assess the significance of the statistical parameters included in the text. In other words, is the standard deviation of the J2+ comparable, larger or smaller with respect to the uncertainties? Please add the missing information and discuss.**

The AIS typically underestimates the ion concentration by 15-30% in the size range of 2-3 nm (Wagner et al., 2016). We had no a separate CPC during the campaign measuring alongside the DMPS to estimate the uncertainty in the DMPS measurement. However, according to Wiedensohler et al. (2012), mobility particle size spectrometers of different design usually have an uncertainty range of around ± 10 % between 20 and 200 nm, and larger uncertainties are expected beyond this size range. In principle, the meteorological measurements at Concordia station follow WMO recommendations. According to WMO 2008 (WMO. Guide to meteorological instruments and

methods of observation. Technical Report 8, World Meteorological Organisation, 2008.), the error for temperature measurement should be < 0.3 K in the range from -40 to +40 C and the uncertainty for pressure measurement should be about 0.1 hPa.

We made an estimation of uncertainties in J2+ by assuming an underestimation of 15-30% in our AIS measurement, an uncertainty of ± 10 % in our DMPS measurement in the whole size range, an error of ± 1 C in the temperature measurement and ± 1 hPa in the pressure measurement. We calculated the maximum and minimum estimates of J2+ based on these assumptions and evaluated the deviations of J2+ from the mean values. For more than 88% of the J2+ (26 cases in total), we obtained a deviation from the mean smaller than 0.020 cm-3s-1 (the standard deviation in all 26 J2+ values). More than 80% of the calculated deviation from the mean has a value smaller than 0.005 cm-3s-1.

*An estimation of uncertainties in $J_2^+$ was made by assuming an underestimation of 15-30% in the AIS measurement (Wagner et al., 2016), an uncertainty of ± 10 % in the DMPS measurement in the whole size range (Wiedensohler et al., 2012), an error of ± 1 C in the temperature measurement and ± 1 hPa in the pressure measurement. We calculated the maximum and minimum estimates of $J_2^+$ based on these assumptions and evaluated the deviations of $J_2^+$ from the mean values of the maximum and minimum estimates. We found that this deviation was smaller than $0.005 \ cm^{-3}s^{-1}$ ($<0.020 \ cm^{-3}s^{-1}$) for more than 80% (88%) of the values of $J_2^+$.*

Minor comments:

Abstracts:
**1) Page1, line 24: "One ... days." Rephrase. It is unclear if the Authors want to refer to event days or event free days.**

We modified the sentence to deliver the message clearer that we refer to days with no sign of new particle formation, wind-induced ion formation, or ion production and loss associated with cloud/fog formation.

*For the subset of days when none of these processes seemed to operate, the concentrations of cluster ions (0.9-1.9 nm) exhibited a clear seasonality, with high concentrations in the warm months and low concentrations in the cold.*

**2) Page 2 Line 4 and 5: The Authors use 2 time the expression "work better" which is very vague. Please rephrase using a more precise terminology.**

The sentence is revised as follows

*The former method seemed to have advantages in characterising NPF events with a fast GR, whereas the latter method is more suitable when the GR was slow.*

**3) Page 2 Line 7 and 8: "The ion production ... seemed a unique feature at Dome C ..." Please be more specific. A unique feature in Dome C relative to the whole world? Relative to the other data collected in Antarctica?**

This refers to a feature that has not been reported for any other sites. The production of ions in the size range of 8-42 nm in relation to cloud formation therefore seems to be a unique feature at Dome C. The sentence is elaborated

*The ion production in relation to cloud/fog formation in the size range of 8-42 nm seemed to be a unique feature at Dome C, which has not been reported elsewhere.*

4**) Page 2 Line 10: "cleavage" replace with a word/rephrase using wording that is understandable to a wide audience.**

We replaced 'cleavage' with 'breakage'.

**3) Page 2 Line 11 and 12 "our observation ... ice crystals." This is a very interesting result! The Referee thinks that it should be explained more thoroughly in the appropriate paragraph (see comment XX)**

We believe that this result was presented and discussed in detail in section 3.3.2. Here in the abstract, we just provided a brief summary.

**Introduction 4) Page 2 line 17: Please add the size/mobility ranges next to "primary ions" and "aerosol particles" for a more complete information for the reader that might be new to atmospheric ions.**

Such information is added to the text.

*Air ions, also known as atmospheric ions, are electric charge carriers in the atmosphere, ranging from primary ions (most likely have a mobility diameter smaller than 0.8-1 nm) to charged aerosol particles (with a mobility diameter up to several hundred nm).*

**5) Page 3 line 2:"Such charged nanoparticles ... are typically observed during new particle formation (NPF) events." Confusing. Stable air ions are observed also without NPF. Please rephrase.**

The reviewer is right. This sentence is modified as
*Charged nanoparticles in the mobility size range of 1.7-7 nm are typically observed during new particle formation (NPF) events.*

**6) Page 3 line 9 to 20: "Carslaw et al. (2013) ... years (Fiebig et al., 2014)." Confusing paragraph. Please add a closing sentence that explains why this work is important and how it is different from the cited previous work in this way the list of previous papers on the topic will make more sense to the reader.**

Sorry for this confusion. This paragraph was meant to go together with the next paragraph. The information the reviewer looks for is actually contained in the following paragraph. These two paragraphs are combined into one in the revised manuscript.

**Methods 7) Page 5 line 4 and 5. Please add the flow rate after "high sample flow rate", which is too vague to be useful. The Referee is aware that there is a flow rate some 20 lines below, but the use of "High" is not helpful and repetita iuvant.**

The detailed flow rate information was given in the third paragraph of section 2.1.1. This first paragraph in section 2.1.1 was meant to serve an introductory purpose for section 2.1.1. However, since the review requested, we add the sample flow rate information as the reviewer suggested.

*The AIS employs two cylindrical multi-channel aspiration-type analysers and a high sample flowrate (60 l/min).*

**8) Page 7 line 21 to 23: The Referee thinks that a plot of the shifted spectrum added to the supplementary material could be very useful to the readers and AIS/NAIS users. I recommend adding it.**

Such a figure is added to the supplementary material.

**9) Page 9 line 4: "growth and coagulation". Remove coagulation.**

Coagulation can cause size increases of an aerosol population. Therefore, coagulation is not removed.

**10) Page 9 line 10: "Air ion and total aerosol particle data are three dimensional: " Remove. It is confusing and unnecessary.**

This sentence is removed as suggested.

**11) Page 9 line 12 and 13: the Authors refer to " mode fitting method" and "appearance time method" throughout as "former" and "latter". Please spell them out each time. Readers will have easier times at understanding to which method the Authors are referring to.**

These methods are spelt out as suggested.

**12) Page 9 line 26: "assist". Confusing term. Please rephrase.**

The word 'assist' is replaced by 'support'.

**13) Page 3 line 3: "Sulphuric acid is considered as a key chemical species". remove "as".**

The sentence is modified as

*Sulphuric acid is a key chemical species in forming aerosol particles in the ambient air.*

**14) Page 11 line 14:"During the campaign period, there were nearly 300 days with valid air ion measurements". Please add what is the corresponding percentage of valid days as well. The Referee is aware that the Authors refer to Table 1 where this information can be retrieve, but thinks that would be of help to the reader to make it explicit in the text.**

This information is given in the text.

*During the campaign period (330 days in total), there were 287 days with valid air ion measurements, i.e. valid air ion data were collected on nearly 87% of the measurement days.*

**15) Page 11 line 18: "definite". Confusing adjective. Please rephrase.**

The word 'definite' is replaced with 'certain'.

**16) Page 12 Line 2: "high" please add a value e.g., median or a range of warm months to help the reader.**

Typically, highest cluster ion concentrations were observed during warm months. Therefore, the sentence is modified as

*The cluster ion concentration was the highest during the warm months, with a maximum in February.*

**17) Page 12 line 4: "natural ionising radiation" is a confusing term as cosmic rays are also natural and ionizing. Please change wording.**

Natural ionizing radiation includes cosmic radiation. See the definition given by IAEA for example https://www.iaea.org/newscenter/multimedia/photoessays/natural-and-artificial-ionizing-radiation

**18) Page 12 line 24: remove "Markedly" it is unnecessary. Please reword the sentence to make it more understandable/intuitive. In addition, do the Authors have information about the boundary layer (BL) in February, March, October and November? Can the Authors exclude that some of those high cluster ion concentration are due to limited mixing due to shallow BL? Please elaborate and include in the text.**

The word 'markedly' is removed as suggested and the sentence is modified.

*The daily-median cluster ion concentration at Dome C was observed to be higher on NPF event days compared with event-free days.*

Unfortunately, we have no information about the boundary layer height and we cannot exclude the shallow BL effect completely. However, the day length in February at Dome C is typically much longer compared with that in March and October. Therefore, a deeper mixed layer can be assumed in February than in March or October. However, the cluster ion concentration was found the highest in February. Therefore, even if the boundary layer height has an effect on the cluster ion concentration, this effect is likely to be very minor. A discussion about the boundary layer height effect is added in section 3.1 as follows

*The development of the planetary boundary layer may additionally influence the concentration of cluster ions by imposing either a dilution or concentration effect. The longer day length in February than in March or October may result in the development of a deeper mixed layer, which could dilute the cluster ions within the mixing volume. However, the highest cluster ion concentration was found in February. Also polar nights would cause the formation of only a very shallow and stable boundary layer in winter months. The mixing volume in winter therefore is expected much smaller than in other seasons, but no concentration effect on cluster ion concentration can be identified. Consequently, even if the seasonal change of boundary layer heights has an influence on the seasonality in cluster ion concentrations, this effect is likely to be minor.*

**19) Page 13 Line 20: "Bumps ... NPF events (Fig. 3c)". Please rephrase being clearer. Consider replacing the word "bumps" with e.g., "sudden increase".**

'bumps' is replaced with 'sudden increases' in the text as suggested by the reviewer.

**20) Page 13 line 24 and 25: "Such differences result ... different origins". Replace "differences" with "different".**

The sentence is modified as

*Such differences result likely from the availability of vapours that sustain the growth.*

**21) Page 13 line 26: replace "perceptible" with "measurable"**

The sentence is modified as

*We could see slight concentration increases in the cluster ion size range at the time when NPF events were initiated, but…*

**22) Page 14 line 11: "Short after". Please be less vague, remove "short" and add temporal information in the text.**

'Short after' is replace by 'About 4 hours after'

**23) Page 14 line 18 to 21: "slowly growing" ... "slight growth". Please add next to this general expression the value of the GR. It will make the text more useful to the reader.**

The growth rate and size range information is added in the text for clearer description. The wording 'slight' was a wrong interpretation. It is corrected in the revised version.

*Over the consecutive five days on 12-16 February, a slowly-growing (GR ≈ 1.4 nm/h) population of 40-200 nm particles could be observed in the background, with their initial formation traceable back to 06:00 UTC on 12 February. Interestingly, apart from the particles initiated at 10 nm and 40 nm, a third mode of particles with sizes larger than 100 nm was recognisable on the morning of 12 February. This particle mode grew approximately from 100 nm to 300 nm during 12-13 February, and then gradually merged with the mode initiated at 40 nm at the end of 16 February.*

**24) Page 14 line 24: Replace "owing to" with "because of"**

We are thankful for he reviewer's suggestion. However, we decide to keep 'owing to' as it is to avoid repeated usage of 'of'.

**25) Page 16 line 19: "... and yields more representative instantaneous GRs". The word "representative" should be reserved to statistical analysis. If such analysis was done to assess whether the calculated GRs were statistically representative please add a sentence about the method used, otherwise rephrase.**

We are afraid that we do not agree with the reviewer. In our opinion, the word 'representative' is not reserved to statistical analysis only. Here we made no statistical evaluation, but we decide to keep the word 'representative'.

**26) Page 16 line 20: The Referee thinks that a comparison between the GR for ions with Dp>10 nm and particles in the same size range should be discussed in this para- graph.**

We add discussion at the end of the last paragraph in section 3.2.2 as the reviewer suggested.

*At large sizes in the overlapping size range (10-42 nm) of AIS and DMPS, the instantaneous GRs derived from the AIS measurements tended to be larger than those from the DMPS measurements. This difference may result from the fact that the DMPS measures total particles, including both ions and neutral particles, whereas the AIS detects only charged particles. Also the AIS measurements at sizes larger than 20 nm are subject to the uncertainties brought by the detection of multiply charged particles as singly charged particles. At small sizes in the overlapping size range, the instantaneous GRs derived from the DMPS exhibited a decreasing trend with increasing sizes, which however was not shown by the instantaneous GRs derived from the AIS. This difference may again be attributed to the difference in the sampled particles targeted by the two instruments.*

**27) Page 17 line 17: "... 110 nm". Based on the data analyzed in this work and on literature is the activation in this size range typical? The Referee recommends, if possible, adding some discussion about this in this paragraph.**

As suggested by the referee, we modified this paragraph into the following form:

*This observation is well in line with the activation thresholds from <50 nm up to about 200-300 nm for the "dry" particle diameter observed in real atmospheric clouds (see Henning et al., 2002, and references therein; Anttila et al., 2009; Kyrö et al., 2013; Portin et al., 2014; Leaitch et al., 2016). Clusters ions were efficiently lost onto the cloud droplets at Dome C.*

**28) Page 18 line 11: "we observed wind-induced ion formation especially during the winter months". Please add some numbers to make this statement less vague. How many times in winter with respect to other months?**

The sentence is modified by taking in to account the reviewer's comment as

*We observed wind-induced ion formation especially during the dark months (15 cases during May-August).*

**29) Page 18 line 24: " Ionising radiation produces primary ions via ionisation". Please either remove "via ionization" or add what is ionized e.g., "via ionization of vapor molecules"**

'via ionisation' is removed according to the reviewer's suggestion.

**30) Page 19 line 8: "However ... contribute to the ion burst captured by the AIS". The Referee thinks that this statement is highly speculative and unsubstantiated. The Authors should discuss more in length adding references.**

The discussion on this point was actually given in the following up text in this paragraph. In this paragraph, we proposed a mechanism that we think might be the cause to the ion formation during strong wind episodes, which needs further experimental validation as mentioned in the text. The text in the beginning of the paragraph is modified to better show the speculative nature of the whole paragraph.

*Turbulent conditions might enhance the collection of electric charges by the shattered snowflakes and ice particles via a charge transfer from initial charge carriers, contributing to the formation of an ion burst. In addition, the shattered particles might gain electric charges through friction charging. However, we think that these two pathways of ion formation are not likely to contribute to the ion burst captured by the AIS. In principle,…*

**31) Page 19 line 22: "similar" the wording makes the sentence too generic and vague, please discuss further, how those feature are similar.**

The sentence is modified as

*By putting together all the 36 wind-induced ion formation events, a linear correlation was identified between the logarithm of the ion concentration and wind speed (Fig. 9), like also found at Aboa.*

**32) Page 20 line 3: "unexpectedly" unnecessary adjective, please remove.**

The word 'unexpectedly' is removed as suggested by the reviewer.

**33) All figures of DMPS and AIS size distributions: add units to dN/dlogDp make the units of Dp consistent (all nm or all m) add thick labels so that are consistent and at least 2 in number e.g., 10 and 100 nm**

units are added to dN/dlogdp and dp are shown in nm and tick labels are added as reviewer suggested to all DMPS and AIS contour plots.

**34) Page 31 caption of figure 1: specify the polarity of the ions, add units to the y-axis, uniform the units of Dp (all nm or all m) and add a tick label near the cluster band.**

The polarity is specified and the unit is added to the y-axis label. Dp are shown in nm and a tick label is added in the contour plot to point on the cluster band. The modified Figure 1 and its captions are

[Figure]

*Figure 1. The median size distribution of* positive *ions measured by the AIS on an event-free day (16 January, 2011). The measured number size distribution of this day is shown in the contour plot.*

**35) Page 32 figure 2: the x-axis is and its label are confusing. Please make the x-axis so that have the same label and take the same space in this way the reader will be able to easily compare CS and cluster ions during the same month.**

Figure 2 is fixed according to the reviewer's suggestion.

[Figure]

Figure 2. Seasonality in the median a) cluster ion (0.9-1.9 nm) concentration and b) condensation sink (CS). Tops and bottoms of the boxes are the 75th and 25th percentiles of the median daily values in 10 min time resolution, with bars in the middle showing the 50th percentiles. Whiskers represent spans of the interquartile ranges multiplied by 1.5. Cluster ion concentrations or CS on new particle formation (NPF) days shown in red and on event-free days in black. Event-free conditions were restricted to days, on which no NPF, cloud activation, wind-induced events or contamination as well as other anomalies altering the ion concentration in the cluster band. The numbers of days classified as either event-free or NPF are displayed on the top of the panel a) in grey colour. No CS was obtained in August due to the lack of measured temperature and pressure data from the station database.

**36) Figure 3, all panels: adding a visual indicator for the fog would help the reader to identify the fog period. Please consider adding it.**

No fog was observed during the time shown in Figure 3.

**37) Figure 3, panel c): The secondary axis have a weird grey halo, please consider fixing it.**

The grey halo is removed.

**38) Figure 5, caption: "envolvement" pleases reword, this might not be English. Maybe "evolution"?**

We used 'evolvement', which is an English word. However, it is changed to 'evolution' as the reviewer suggested.

**39) Figure 7, panel f): The Referee thinks that adding a label "D50 = 1.1e-7 m", or even better "D50 = 110 nm" instead of the number alone would make the figure easier to read.**

Figure 7f is modified according to the reviewer's suggestion.

[Figure]

**40) Figure 9 and S3, all panels: please give only the significant digits for the fit.**

The fitting parameters are rounded to 2 significant digits and are presented in Table S1 in the revised manuscript.

*Table S1. Coefficients for the fittings shown in Figs. 9 and S4. $R^2$ is the coefficient of determination measuring the goodness of fit, which denotes the fraction of the total variation in the data can be explained by the fit. For Dome C data shown in Fig. 9, fits 1 and 2 are obtained based on all data below or above the wind speed threshold (7 m/s), respectively. The grey data points in Fig. 9 are used in determining the fitting coefficients for fits 3 and 4. For Aboa data shown in Fig. S4, a wind speed threshold of 17 m/s is used.*

| | | | | Cluster (0.9-1.9 nm) ion concentrations vs. wind speeds | | |
|---|---|---|---|---|---|---|
| | *Fits* | *a* | *b* | *95% conference interval for a* | *95% conference interval for b* | $R^2$ |
| *DOME C (Fig. 9)* | *1* | *0.69* | *26.34* | *[0.65 0.73]* | *[21.62 32.10]* | *0.24* |
| | *2* | *0.51* | *68.64* | *[0.41 0.60]* | *[29.88 157.67]* | *0.12* |
| | *3* | *0.73* | *21.83* | *[0.69 0.77]* | *[18.02 26.44]* | *0.28* |
| | *4* | *0.44* | *327* | *[0.40 0.47]* | *[244.95 436.53]* | *0.52* |
| | *1.9-10 nm ion concentrations vs. wind speeds* | | | | | |

| | Fits | a | b | 95% conference interval for a | 95% conference interval for b | $R^2$ |
|---|---|---|---|---|---|---|
| | 1 | 1.14 | 0.07 | [1.08 1.2] | [0.05 0.09] | 0.29 |
| | 2 | 0.61 | 2.1 | [0.71 0.88] | [0.88 5.01] | 0.15 |
| | 3 | 1.19 | 0.05 | [1.25 0.04] | [0.04 0.07] | 0.31 |
| | 4 | 0.54 | 9.87 | [0.58 7.18] | [7.18 13.58] | 0.58 |
| *ABOA (Fig. S4)* | *0.9-2.2 nm ion concentrations vs. wind speeds* | | | | | |
| | Fits | a | b | 95% conference interval for a | 95% conference interval for b | $R^2$ |
| | 6 | 0.17 | 14.98 | [0.15 0.19] | [9.02 24 88] | 0.66 |
| | *2.2-9.5 nm ion concentrations vs. wind speeds* | | | | | |
| | Fits | a | b | 95% conference interval for a | 95% conference interval for b | $R^2$ |
| | 5 | 0.24 | 1.45 | [0.22 0.26] | [1.26 1.68] | 0.35 |
| | 6 | 0.06 | 63.24 | [0.04 0.09] | [34.72 115.18] | 0.17 |

References:

[revised manuscript text omitted]